# SplineGS: Learning Smooth Trajectories in Gaussian Splatting for Dynamic Scene Reconstruction

**Jihwan Yoon**[1]**, Sangbeom Han**[1]**, Jaeseok Oh**[1]**, Minsik Lee**[1,2*]
[1]Department of Electrical and Electronic Engineering, Hanyang University
[2]School of Electrical Engineering, Hanyang University ERICA
{yunwlghks, gkstkdqja88, oh07020, mleepaper}@hanyang.ac.kr

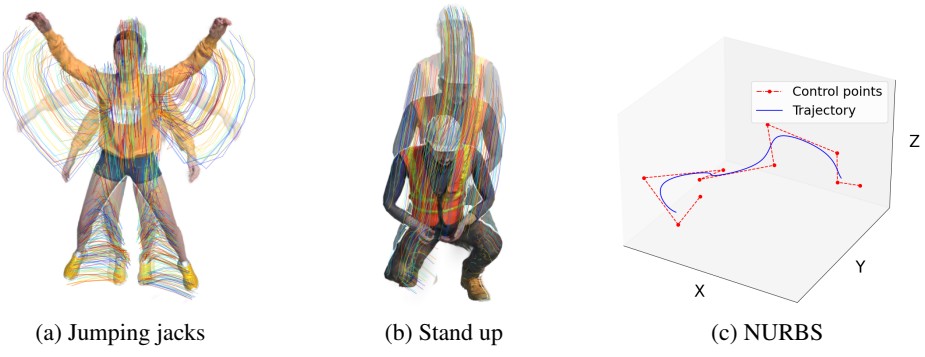

| (a) Jumping jacks | (b) Stand up | (c) NURBS |

Figure 1: We propose SplineGS, which represents the trajectories in Gaussian splatting using non-uniform rational B-spline (NURBS) to reconstruct dynamic scenes. The learned trajectories are shown with rendering results in (a) and (b). (c) shows an example of a trajectory based on NURBS.

## Abstract

Reconstructing complex scenes with deforming objects for novel view synthesis is a challenging task. Recent works have addressed this with 3D Gaussian Splatting, which effectively reconstructs static scenes with high quality in short training time, by adding specialized modules for the deformations of Gaussian blobs. However, designing an effective deformation module that incorporates appropriate spatiotemporal inductive biases still remains unresolved. To address this issue, we propose SplineGS in this paper, which utilizes non-uniform rational B-splines (NURBS), an extension of B-spline, to represent temporally smooth deformation. A set of representative trajectories are learned based on NURBS, and the individual trajectories of Gaussian blobs are represented as linear combinations of these trajectories for spatial smoothness. The weights of the combinations are trained based on a multi-resolution hash table and an MLP, with the positions of the Gaussian blobs as the keys. Thanks to this design, the proposed method does not need any regularizers for trajectories, which enables efficient training. Experiments demonstrate that the proposed method provides competitive performance over the existing methods with much shorter training time.

## 1 Introduction

3D reconstruction is a fundamental problem in computer vision and graphics that has been studied for decades. Recently, neural radiance fields (NeRF) (Mildenhall et al., 2020) have achieved a remarkable breakthrough in novel view synthesis (NVS) based on implicit representations. However, NeRF still suffered from limited performance and, especially, slow training times, prompting the development of various subsequent methods (Yu et al., 2021; Fridovich-Keil et al., 2022; Chen et al.,

---
*Corresponding author

2022; Barron et al., 2021; 2023). Recently, 3D Gaussian Splatting (3D-GS) (Kerbl et al., 2023) has emerged as a successful alternative by representing scenes based on 3D Gaussian blobs and employing a differentiable tile rasterizer, resulting in fast, high-quality rendering and thus reduced training time.

Despite the success of 3D-GS in static scenes, applying it to dynamic scenes with deforming objects remains a challenging task due to the difficulty in modeling effective trajectory representations. Recent works have proposed various methods to model the trajectories of 3D Gaussian blobs over time to reconstruct dynamic scenes. Approaches that expresses deformations based on neural representations (Yang et al., 2024b; Wu et al., 2024) optimize canonical 3D Gaussian blobs as well as a neural module to predict temporal deformations. Neural representations in these approaches, however, do not provide enough inductive biases for deformations, and additional training mechanisms or smoothness regularizers are utilized to handle these. Another approach directly optimizes 4D Gaussian blobs (Yang et al., 2024a; Duan et al., 2024), which introduces challenges in optimization due to the higher dimensionality (4D). This results in reconstruction artifacts, such as floating objects that do not align correctly with the scene. Even though these previous works have shown impressive performance, the trajectory representations they use impose some limitations on either performance, convergence speed, or both.

In this paper, we propose a novel method, i.e., SplineGS, that models the trajectories of 3D Gaussian blobs as temporally smooth representations based on non-uniform rational B-splines (NURBS) (Piegl & Tiller, 2012), an extension of B-splines (De Boor, 2001), to reconstruct dynamic scenes. NURBS can represent continuous and smooth trajectories, similar to B-splines, while allowing for more flexible and finer adjustments. Our approach specifically obtains a set of representative trajectories for both positions and rotations through NURBS, of which the parameters are learned during the training. To ensure spatial smoothness, the individual trajectory of each Gaussian blob is expressed as a linear combination of these representative trajectories. Inspired by Müller et al. (2022), the weights for these combinations are obtained using a multi-resolution hash table and a multi-layer perceptron (MLP), where the average position of each Gaussian blob serves as the key. The NURBS representation and the linear combination strategy act as implicit regularizers embedded within the structure. Based on this smooth representation, our approach achieves high-fidelity rendering and fast training time without relying on any explicit smoothness regularizers, making it efficient for complex dynamic scenes.

We evaluated the proposed method on two dynamic-scene datasets, i.e., D-NeRF (Pumarola et al., 2021) and Neu3D (Li et al., 2022). Experimental results demonstrate that the proposed method achieves highly competitive performance, even with shorter training time.

Our contributions are summarized as follows:

- We propose temporally smooth representations for the trajectories of Gaussian blobs based on NURBS.

- We enforce spatial smoothness by introducing a low-rank assumption for the trajectories. Each Gaussian blob's trajectory is modeled as a weighted sum of representative trajectories, where the weights are learned by a multi-resolution hash table and an MLP. This eliminates the need for a separate smoothness regularizer.

- The proposed smooth representation exhibits physically plausible trajectories, which leads to high-fidelity rendering and fast training times.

## 2 RELATED WORK

In this section, we briefly review the NeRF-based methods for dynamic scenes. Following this, we introduce the original 3D-GS for static scenes and the methods that extend 3D-GS to dynamic scenes.

### 2.1 NEURAL RADIANCE FIELDS FOR DYNAMIC SCENES

In recent years, NeRF-based methods have gained significant attention for solving the NVS problem of static scenes. This surge in interest was largely due to the impressive performance of vanilla

NeRF, which has inspired subsequent research efforts aimed at improving memory usage, training speed, rendering quality, and rendering speed. In addition, many attempts have been made to extend it to dynamic scenes. D-NeRF (Pumarola et al., 2021) and Nerfies (Park et al., 2021a) utilize the coordinates in a canonical space and time embedding vectors as inputs to the MLP to compute the deformations. HyperNeRF (Park et al., 2021b) further enhanced this approach by embedding a template NeRF in a higher-dimensional space to better capture topological changes. However, these methods often suffer from inefficiency in rendering due to the high number of queries required for the MLP. Spline-NeRF (Knodt, 2022) takes all sampled points along a ray as inputs to an MLP, which then outputs control points for Bézier curves. However, it is difficult to represent a complex trajectory, e.g., one that has many temporally local changes, based on a Bézier curve. A Bézier curve is a global representation, i.e., changes in some control points affect the entire shape of the curve. Accordingly, it becomes challenging to apply this method to long video sequences. On the other hand, $K$-Planes (Fridovich-Keil et al., 2023) and HexPlane (Cao & Johnson, 2023) represented 4D spacetime by combining multiple 2D planes, improving speed and interpretability through explicit representations. Nevertheless, the complexity of dynamic scenes and the inherent limitations of ray-casting-based rendering prevent these methods from achieving high-fidelity and high-speed rendering required for practical applications.

## 2.2 3D Gaussian splatting for dynamic scenes

Recently, 3D-GS has emerged as a successful solution for reconstructing static scenes, achieving high-fidelity rendering, fast rendering speed, and fast training time. 3D-GS directly optimizes the means, scales, rotations, opacities, and colors of 3D Gaussian blobs that compose a static scene using a differentiable tile-based rasterizer. Inspired by this success, research efforts have begun to explore the use of 3D-GS for reconstructing dynamic scenes (Wu et al., 2024; Yang et al., 2024b; Huang et al., 2024; Yang et al., 2024a; Duan et al., 2024; Luiten et al., 2024; Bae et al., 2024). For example, 4D-GS (Wu et al., 2024) introduces a spatial-temporal structure encoder, which is composed of a multi-resolution HexPlane and a tiny MLP, along with additional MLPs to compute the deformations. In another study (Yang et al., 2024b), purely implicit networks, i.e., MLPs, are used to compute temporal deformations. In SC-GS (Huang et al., 2024), anchor-based spatial warping and MLPs are learned to represent deformations of the overall 3D Gaussian blobs. However, these approaches rely on neural representations, which do not provide enough inductive biases, so additional training mechanisms or smoothness regularizers are utilized. On the other hand, there are methods (Yang et al., 2024a; Duan et al., 2024) that directly optimize 4D Gaussian blobs. While this approach allows for a flexible representation of complex dynamic scenes, it also introduces challenges in optimization due to the increased dimensionality (4D). These challenges can lead to reconstruction artifacts, such as floating objects, and achieving high-quality reconstructions requires additional regularizers or extended training time. Alternatively, polynomial and Fourier bases (Lin et al., 2024) can also be used to represent trajectories. However, since this approach must utilize all the bases to calculate deformation at a specific time, it can be more challenging to model temporally local deformations and the computational cost can increase significantly when reconstructing long video sequences.

## 3 The proposed method

In this section, we propose SplineGS, which represents deformations based on spatiotemporally smooth representations. First, we review 3D Gaussian Splatting in Section 3.1. In Section 3.2, we introduce the B-spline (non-uniform rational B-splines (NURBS), specifically) representations for the trajectories of Gaussian blobs. Then, in Section 3.3, a multi-resolution hash table and an MLP are utilized to enforce spatial smoothness based on linear combinations of representative trajectories. Finally, in Section 3.4, we describe the objective function for optimization. The pipeline of SplineGS is illustrated in Figure 2.

### 3.1 Preliminary: 3D Gaussian splatting

3D-GS Kerbl et al. (2023) reconstructs a static scene by directly optimizing the means, covariance matrices, opacities, and colors of 3D Gaussian blobs. Given $N$ Gaussian blobs, the shape of each

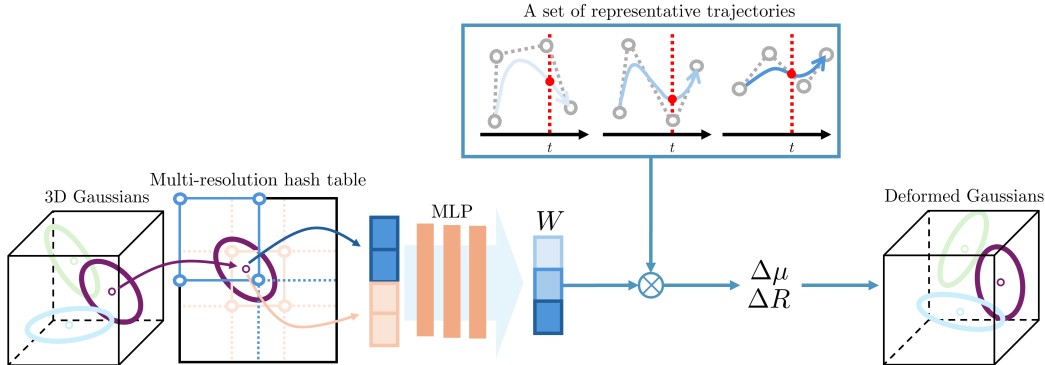

Figure 2: **The pipeline of the proposed method.** For each given 3D Gaussian blob, the positions normalized into the range of $[0, 1]^3$ are used as keys for the multi-resolution hash table. The features derived from the position are then mapped to weights via an MLP for the linear combination of representative trajectories. Meanwhile, the relative positions and rotations of the representative trajectories at time $t$ are calculated based on NURBS. Finally, the deformation of each Gaussian blob is obtained by a linear combination.

Gaussian is represented by its mean $x_i$ and covariance matrix $\Sigma_i$:

$$G_i(\mu) = e^{-\frac{1}{2}(\mu - \mu_i)^T \Sigma_i^{-1} (\mu - \mu_i)}, \tag{1}$$

The covariance matrix must be positive semi-definite, and this is ensured during optimization by separately optimizing the corresponding scales and rotations, i.e., $S_i$ and $R_i$:

$$\Sigma_i = R_i S_i S_i^T R_i^T. \tag{2}$$

To render the 3D Gaussian blobs in the scene onto a 2D image space, the 2D covariance matrix $\Sigma_i'$ in camera coordinates is derived using a given camera extrinsic transform $W$ and the Jacobian of the affine approximation matrix $J$ (Zwicker et al., 2001):

$$\Sigma_i' = JW\Sigma_i W^T J. \tag{3}$$

Afterward, the pixel's color $C$ is computed by blending $N'$-ordered blobs that overlap the pixel.

$$C = \sum_{k=1}^{N'} c_{i_k} \alpha_{i_k} \prod_{j=1}^{k-1} (1 - \alpha_{i_j}). \tag{4}$$

where $\{i_k\}$ represents the indices of the overlapping blobs and $c_i$ denotes the color of each blob. $\alpha_i$ is determined by evaluating the 2D Gaussian with covariance $\Sigma_i'$ and multiplying it by a learned per-point opacity.

### 3.2 Trajectory representations based on B-splines

We employ non-uniform rational B-splines (NURBS) (Piegl & Tiller, 2012) to represent smooth trajectories over time. Here, we first explain the basic formulation based on B-splines, and then extend it to NURBS.

**B-splines.** B-splines (De Boor, 2001) are defined as a set of piecewise polynomial functions, where each polynomial has a specific degree denoted as $p$. These polynomial functions are connected continuously at specific points called knots, represented by $t_0, t_1, \ldots, t_{p+M+1}$. Here, $M$ is the number of control points which will be defined later. The curve's shape is determined as a linear combination of control points and their corresponding basis functions. When $p = 0$, the influence of the $i$-th control point at a given time is described by the basis function as follows:

$$B_{i,0}(t) := \begin{cases} 1 & \text{if } t_i \leq t < t_{i+1}, \\ 0 & \text{otherwise.} \end{cases} \tag{5}$$

The higher-degree B-splines, starting from $p = 0$, can be defined using the Cox-de Boor recursion formula:

$$B_{i,p}(t) := \frac{t - t_i}{t_{i+p} - t_i} B_{i,p-1}(t) + \frac{t_{i+p+1} - t}{t_{i+p+1} - t_{i+1}} B_{i+1,p-1}(t). \tag{6}$$

Once the basis functions are determined, the position at any given time $t$ is computed as a linear combination of the control points, with these functions acting as weights.

$$S(t, P) = \sum_{i=0}^{n} B_{i,p}(t) P_i \tag{7}$$

where $S(t, P)$ denotes the position of the B-spline curve at time $t$ and $P = \{P_i\}$ are the control points that define the curve's shape.

**Non-uniform rational B-splines.** To represent more complex trajectories, the trajectory of a Gaussian blob can be modeled using non-uniform rational B-spline (NURBS), an extension of B-splines. NURBS allows for finer controls of the trajectory by incorporating an additional weight for each control point, and a NURBS curve is expressed as follows:

$$S(t, w, P) = \frac{\sum_{i=0}^{M} B_{i,p}(t) w_i P_i}{\sum_{i=0}^{M} B_{i,p}(t) w_i} \tag{8}$$

where $w_i$ denotes the weight associated with the control point $P_i$. Here, assigning uniform weights makes it equivalent to plain B-splines. Assigning non-uniform weights allows for more flexible adjustment over the control points that represent complex motion. Additionally, during simpler motions, simplification can be encouraged by adopting low weights.

In this paper, we utilize NURBS to represent the trajectory of a Gaussian blob (i.e., translation and rotation), and $P$ and $w_i$ are regarded as learnable parameters. Note that the NURBS trajectories in this paper are defined as the differences from the canonical (or static) positions or rotations. In other words, the trajectories must be combined with the static positions or rotations before use. The static positions and rotations of blobs are separate learnable parameters in the proposed method, along with colors defined as spherical harmonics, opacities, and scales. The degree $p$ and the number of control points $M$ are hyperparameters, and the number of knots is determined as $p + M + 1$. The locations of knots are defined as equally spaced points in the entire time frame.

## 3.3 BLENDING REPRESENTATIVE TRAJECTORIES FOR SPATIAL SMOOTHNESS

Optimizing B-splines for each 3D Gaussian blob separately has several drawbacks: First of all, it requires a large memory space and can increase the overall complexity of optimization. More importantly, spatial smoothness, which is expected in between the trajectories in proximity, is not guaranteed. To resolve these issues all at once, we instead represent the individual trajectories as weighted sums of a few representative trajectories, which only are defined based on NURBS. The weights for these sums are obtained by a multi-resolution hash table and an MLP, which is inspired by instantNGP (Müller et al., 2022), using the position of each Gaussian blob as the key:

$$\beta_j = \tanh(\text{MLP}(\text{Hash}(\mu_j))) \quad \text{for } j = 1, 2, \ldots, N \tag{9}$$

where $\mu_j \in \mathbb{R}^3$ represents the center position of the $j$-th blob, $N$ denotes the number of Gaussians, $\beta_j \in \mathbb{R}^L$ is the weight vector of the corresponding Gaussian blob, and $L$ is the number of representative trajectories. We apply $\tanh$ to the weights to limit their range to $[-1, 1]$. Note here that we do not use any explicit spatial-proximity-based operations, such as kernel regression or $K$-nearest neighbors (Huang et al., 2024), and instead learn the weights implicitly by the hash table and MLP. This can effectively prevent the overall computational complexity of the training procedure from increasing.

After obtaining the weights, the positions and rotations of the representative trajectories at time $t$, calculated based on NURBS, are combined by the weights. The weighted sum operation is applied in the original 3D space for the positions, while it is applied to the axis-angle representations for the rotations and then converted to quaternions.

$$\Delta\mu_j(t) = \sum_{k=0}^{L-1} \beta_{j,k} \cdot S(t, w_k, P_k^{\text{pos}}), \tag{10}$$

$$\Delta R_j(t) = \mathcal{Q}\left(\sum_{k=0}^{L-1} \beta_{j,k} \cdot S(t, w_k, P_k^{\text{rot}})\right), \quad (11)$$

$$\mu_j(t) = \mu_j + \Delta\mu_j(t), \quad (12)$$

$$R_j(t) = R_j \cdot \Delta R_j(t), \quad (13)$$

where $P_k^{\text{pos}} \in \mathbb{R}^{M \times 3}$ and $P_k^{\text{rot}} \in \mathbb{R}^{M \times 3}$ represent the control points for positions and rotations, respectively, for the $k$-th representative trajectory, and $w_k \in \mathbb{R}^M$ denotes the corresponding weights (of the control points) in NURBS. $\mathcal{Q}$ denotes the axis-angle-to-quaternion conversion. The position and rotation from the first two equations are relative ones (from the static position and rotation, respectively), so they are combined with the static ones in the last two equations. The above technique effectively suppresses unwanted, abrupt spatial variations and promotes gradual changes. Accordingly, the proposed method can ensure spatial smoothness without increasing the overall computational complexity, and also has compact feature representations.

## 3.4 OPTIMIZATION

The proposed method incorporates the above spline module on top of the original 3D-GS to estimate deformations. This process is performed end-to-end, and we term this method SplineGS.

**Reconstruction loss.** The reconstruction loss is defined as the difference between the ground truth image and the rendered image. Similar to 3D-GS (Kerbl et al., 2023), we include both $\mathcal{L}_1$ and $\mathcal{L}_{\text{D-SSIM}}$:

$$\mathcal{L}_{\text{recon}} = (1 - \lambda)\mathcal{L}_1 + \lambda\mathcal{L}_{\text{D-SSIM}} \quad (14)$$

where $\lambda$ is a predefined weight.

**Sparsity.** In complex scenes, particularly with real-world datasets, the diversity and complexity of object structures lead to an increased number of Gaussian blobs to represent fine details. This, on the other hand, generates artifacts, such as floating objects, and increases memory usage. To prevent the creation of unnecessary 3D Gaussian blobs, we add L1 regularization on the opacity values:

$$\mathcal{L} = \mathcal{L}_{\text{recon}} + \lambda_o \sum_{j=1}^{N} |o_j| \quad (15)$$

where $\lambda_o$ is also a predefined weight and $o_j$ is the opacity of the $j$th blob.

## 4 EXPERIMENTS

In this section, we evaluate SplineGS on the D-NeRF (Pumarola et al., 2021) and Neu3D (Li et al., 2022) datasets. Performance was measured using PSNR, D-SSIM, training time, and FPS, and compared with existing NeRF-based and 3D-GS-based methods.

### 4.1 DATASETS

**D-NeRF dataset.** The D-NeRF dataset consists of monocular video frames with well-aligned camera poses. This synthetic dataset includes 8 scenes: *bouncingballs, hellwarrior, hook, jumpingjacks, mutant, standup, trex* and *lego*, each of which contains between 50 and 200 training images and 20 test images. The experiments were conducted at a resolution of 800×800.

**Neu3D dataset.** This is a real-world dataset that includes 6 scenes: *coffee_martini, cook_spinach, cut_roasted_beef, sear_steak, flame_steak* and *flame_salmon*. Each scene, except for *flame_salmon*, was recorded for 10 seconds at 30 fps using 15 to 20 fixed cameras, with one camera designated as the test view. The experiments were conducted at half resolution, i.e., 1352×1014. We did not use *flame_salmon*, which consists of 1200 frames, in the experiments, following the practices of many existing methods (Guo et al., 2024; Liu et al., 2024; Lu et al., 2024).

Table 1: **Quantitative comparison for the D-NeRF dataset.** We compared our method to existing methods on 800×800 resolution test images. Here, "40K" and "80K" in the parentheses indicate the numbers of training iterations. The asterisks (*) indicate that the results were adopted from the 4D-GS paper (Wu et al., 2024), while the daggers (†) indicate that those were from the original papers. All the other results were reproduced in our experiments. The average PSNR and SSIM are measured across all scenes, with some cells highlighted to represent the best , second best , and third best . Additionally, we report the average training time and rendering speed in frames per second (FPS) for each method.

| Method | PSNR↑ | SSIM↑ | Training time↓ | FPS↑ |
|---|---|---|---|---|
| $K$-Planes* (Fridovich-Keil et al., 2023) | 30.67 | 0.9672 | 52 mins | 0.97 |
| HexPlane* (Cao & Johnson, 2023) | 31.02 | 0.9680 | 11 mins | 2.5 |
| TiNeuVox* (Fang et al., 2022) | 31.35 | 0.9613 | 28 mins | 1.5 |
| D-3DGS.† (Yang et al., 2024b) | 38.50 | 0.9857 | 22 mins | 70 |
| 4D-GS* (Wu et al., 2024) | 34.06 | 0.9787 | 13 mins | 62 |
| 4DGS (Yang et al., 2024a) | 32.87 | 0.9649 | 7.5 hours | 135 |
| CoGS† (Yu et al., 2024) | 37.90 | 0.9842 | - | - |
| CompDynGS† (Katsumata et al., 2024) | 33.21 | 0.9770 | 8 mins | 150 |
| SC-GS (Huang et al., 2024) | 39.53 | 0.9906 | 1.1 hours | 111 |
| SplineGS (40K) | 39.11 | 0.9866 | 17 mins | 188 |
| SplineGS (80K) | 39.44 | 0.9868 | 35 mins | 188 |

4D-GS (Wu et al., 2024)   4DGS (Yang et al., 2024a) D-3DGS (Yang et al., 2024b)      SplineGS            GT

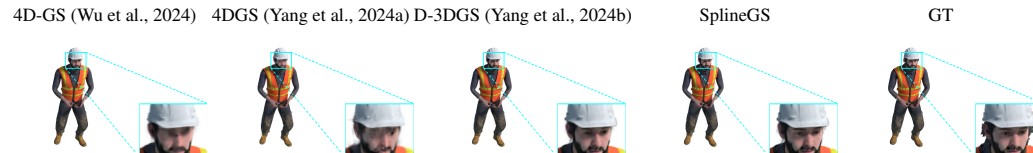

Figure 3: **Qualitative results for the D-NeRF dataset.** We present the rendering results of various methods, including ours, with the ground truth (GT) for *standup* scene in the D-NeRF dataset. Additional rendering results for other scenes are provided in the appendix.

## 4.2 IMPLEMENTATION DETAILS

We implemented SplineGS by integrating the proposed spline-based deformation module, implemented using PyTorch (Paszke et al., 2019), with the original 3D-GS (Kerbl et al., 2023). For the D-NeRF dataset, we randomly sample 100,000 points within a 3D space, where each axis ranges from -1.3 to 1.3, to initialize 3D Gaussian blobs. For the Neu3D dataset, the initial set of 3D Gaussians is obtained by calculating structure-from-motion (Schönberger & Frahm, 2016) for the first frames of each video and downsampling the results. The detailed settings for this were identical to those in 4D-GS (Wu et al., 2024). The initialization of scales, rotations, colors, opacities, and their respective learning rates follows the 3D-GS. Similarly, during training, we adopt the clone and split strategies with a threshold of $2 \times 10^{-4}$, as in 3D-GS. For the D-NeRF dataset, we also utilized the repeated opacity reset suggested in 3D-GS. The training process begins with initial 3K iterations of static 3D-GS training (ignoring any deformations), followed by an end-to-end training with the proposed spline module.

For the spline module, the number of control points (per trajectory) was set as one for every four to eight frames collected from a single camera. These control points were initialized with a normal distribution with a standard deviation $10^{-5}$. The number of representative trajectories was set to either 64 or 128, depending on the dataset. The initial learning rate for the control points was set to $10^{-3}$ for the D-NeRF dataset and $5 \times 10^{-3}$ for the Neu3D dataset, with a decay factor of 0.99 for every 100 iterations. Similarly, the learning rate of $w_k$ was initialized to $10^{-3}$ for the D-NeRF dataset and $5 \times 10^{-4}$ for the Neu3D dataset, and also decayed by a factor of 0.99 for every 100 iterations. The multi-resolution hash table and MLP were implemented using *tiny-cuda-nn* (Müller, 2021), with 16 levels and 4 features per level. The maximum table size parameter, base resolution,

Table 2: **Quantitative comparison for the Neu3D dataset.** We compared our method with existing methods on 1352×1014 resolution test images. Here, "20K" and "40K" in the parentheses indicate the numbers of training iterations. The conventions for typographical marks and color codes are generally identical to Table 1. We measured the average PSNR and SSIM across all scenes except *flame_salmon*, following the practice of many existing works (Guo et al., 2024; Liu et al., 2024; Lu et al., 2024). The double dagger (‡) denotes that the mean values were calculated across all scenes in the respective work.

| Method | PSNR↑ | SSIM↑ | Training time↓ | FPS↑ |
|---|---|---|---|---|
| NeRFPlayer[*] (Song et al., 2023) | 32.12 | 0.9206 | 5.5 hours | 0.045 |
| K-Planes[*] (Fridovich-Keil et al., 2023) | 31.86 | 0.9658 | 1.8 hours | 0.23 |
| 4D-GS[*] (Wu et al., 2024) | 31.54 | 0.9444 | 40 mins | 30 |
| Gaussian-Flow[‡] (Lin et al., 2024) | 32.00 | 0.9700 | 42 mins | - |
| 4DGS[†] (Yang et al., 2024a) | 32.53 | - | - | 114 |
| STG[†] (Li et al., 2024) | 32.57 | 0.9740 | 42 mins×6 | 140 |
| SplineGS (20K) | 32.52 | 0.9484 | 27 mins | 76 |
| SplineGS (40K) | 32.60 | 0.9496 | 55 mins | 76 |

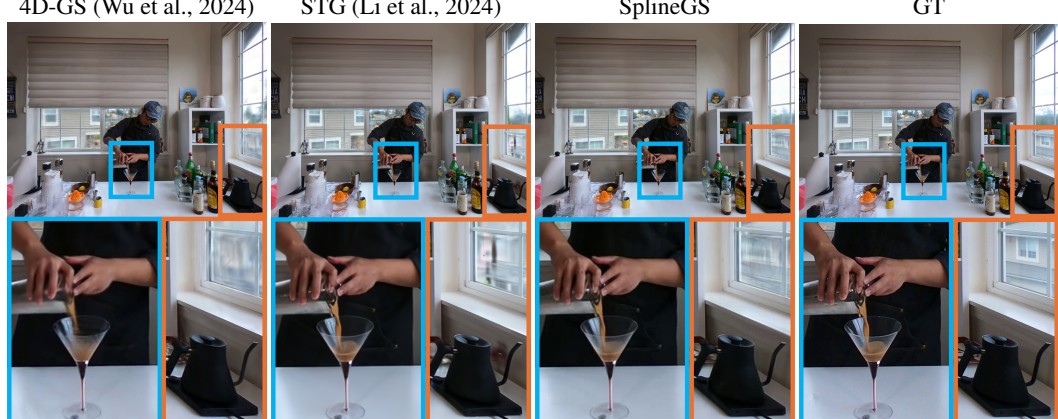

| 4D-GS (Wu et al., 2024) | STG (Li et al., 2024) | SplineGS | GT |
|---|---|---|---|

Figure 4: **Qualitative results for the Neu3D dataset.** We present the rendering results of various methods, including ours, and the ground truth (GT) for *coffee_martini* in the Neu3D dataset. We also provide zoomed-in results in the second row of the figure. Additional results for other scenes are provided in the appendix.

and per-level scale were set to 19, 8, and 2, respectively, for the D-NeRF dataset and 15, 16, and 1.5, respectively, for the Neu3D dataset. The MLP consisted of 4 hidden layers with hidden dimensions of 128 for all the layers. The optimization of the spline module, like 3D-GS, was performed using Adam (Kingma, 2014).

The performance of SplineGS was evaluated after 40K and 80K iterations on the D-NeRF dataset, and 20K and 40K iterations on the DyNeRF dataset, with results averaged over three runs. All the experiments were conducted on a single Nvidia Geforce RTX 3090.

## 4.3 RESULTS

**Results on synthetic scenes.** Here, we compared the proposed model with various NeRF-based methods: K-Planes (Fridovich-Keil et al., 2023), HexPlane (Cao & Johnson, 2023), and TiNeuVox (Fang et al., 2022); as well as 3D-GS-based methods: Yang et al. (2024b), 4D-GS (Wu et al., 2024), 4DGS (Yang et al., 2024a), CoGS (Yu et al., 2024), Katsumata et al. (2024) and SC-GS (Huang et al., 2024)) on the D-NeRF dataset. A quantitative evaluation of average PSNR, SSIM, training time, and FPS is shown in Table 1. Here, the proposed method ranks second best, but with significantly shorter training time compared to the best performing one while having a very small performance gap.

Table 3: **Effect of the degree of NURBS.** We quantitatively evaluated the effect of the degree of NURBS in our method on synthetic and real-world scenes.

| Degree $p$ | Bouncing Balls | | Hook | | Coffee Martini | | Sear Steak | |
|---|---|---|---|---|---|---|---|---|
| | PSNR↑ | SSIM↑ | PSNR↑ | SSIM↑ | PSNR↑ | SSIM↑ | PSNR↑ | SSIM↑ |
| $p = 0$ | 36.03 | 0.9915 | 28.43 | 0.9441 | 28.42 | 0.9127 | 33.61 | 0.9606 |
| $p = 1$ | 40.93 | 0.9951 | 38.09 | 0.9879 | 28.56 | 0.9154 | 33.49 | 0.9610 |
| $p = 2$ | 41.26 | 0.9954 | 38.59 | 0.9890 | 29.01 | 0.9178 | 33.66 | 0.9621 |
| $p = 3$ | 41.81 | 0.9956 | 38.71 | 0.9891 | 29.26 | 0.9189 | 33.82 | 0.9624 |
| $p = 4$ | 39.97 | 0.9946 | 26.98 | 0.9311 | 28.64 | 0.9184 | 33.59 | 0.9621 |

Table 4: **Ablation study.** We quantitatively evaluated the effect of different components of the proposed method on synthetic and real-world scenes.

| Method | Bouncing Balls | | Hook | | Coffee Martini | | Sear Steak | |
|---|---|---|---|---|---|---|---|---|
| | PSNR↑ | SSIM↑ | PSNR↑ | SSIM↑ | PSNR↑ | SSIM↑ | PSNR↑ | SSIM↑ |
| w/o hash | 41.20 | 0.9949 | 36.39 | 0.9813 | 27.98 | 0.9041 | 30.87 | 0.9473 |
| w/o MLP | 38.33 | 0.9927 | 37.80 | 0.9879 | 28.80 | 0.9160 | 33.19 | 0.9605 |
| w/o NURBS | 39.83 | 0.9944 | 26.82 | 0.9306 | 28.85 | 0.9181 | 32.08 | 0.9559 |
| proposed | 41.81 | 0.9956 | 38.71 | 0.9891 | 29.26 | 0.9189 | 33.82 | 0.9624 |

This observation suggests that the proposed spline representation is well-suited for dynamic scene reconstruction. More detailed qualitative and quantitative comparisons for each scene are provided in the appendix. Figure 3 shows the qualitative comparison for *standup*. Here, we can confirm that the proposed method yields more accurate rendering compared to the existing methods.

**Results on Neu3D.** For the Neu3D dataset, we compared with the following NeRF-based methods: NeRFPlayer (Song et al., 2023), $K$-Planes Fridovich-Keil et al. (2023); as well as 3D-GS-based methods: 4D-GS (Wu et al., 2024), 4DGS (Yang et al., 2024a) and STG (Li et al., 2024). Table 2 presents the quantitative comparison. For this dataset, the proposed method achieves the best average PSNR after 40K iterations. Moreover, the proposed method always achieves better performance per training time than the existing methods. Here, STG divides the entire video frames (300 frames) into 6 parts (50 frames each) and then processes them separately, which is why the training time is described as "42 mins × 6". Again, the detailed results for each scene are provided in the appendix. Figure 4 shows the qualitative comparisons on the *coffee_martini* scene. Here, the proposed method provides crisper details for the *coffee_martini*.

SplineGS concentrates on the temporal changes of position and rotation, without modeling those of density and color unlike some recent methods (Yang et al., 2024a; Li et al., 2024; Lin et al., 2024). Nevertheless, it achieves high-fidelity rendering and fast convergence, which suggests that the proposed spline representation provides a better alternative for modeling deformations.

## 4.4 ANALYSES AND ABLATION STUDY

**Effect of the degree of NURBS.** In NURBS, the value of a trajectory at a specific time instance is influenced by surrounding $p + 1$ control points. Accordingly, as the degree increases, the curve becomes less sensitive to the changes in control points, which makes the trajectory smoother but also increases the computational cost. Table 3 shows the performance variations according to the changes in the degree for *bouncing balls, hook, coffee_martini, sear_steak*. The results indicate that $p = 3$ generally provides optimal performance for dynamic scenes.

**Ablation study on the module components.** We provide an ablation study on the spline module in Table 4. Here, "w/o hash" indicates that the multiresolution hash table was replaced with a positional encoding $\gamma(p) = \left( \sin \left( 2^k \pi p \right), \cos \left( 2^k \pi p \right) \right)_{k=0}^{L-1}$ with $L = 10$. In this case, performance generally decreases except for *bouncingballs*, which contains less complex motions. In the case of "w/o MLP", the hash table directly outputted the weights for the linear combination, which generally resulted in lower performance. In the "w/o NURBS" scenario, the MLP directly outputs the values

of a trajectory given a time instance $t$: the features generated by the hash table are concatenated with the positional encoding (with $L = 6$) of $t$, and then go through the MLP to output the values. This variation leads to lower performance, which suggests that the neural representation alone lacks sufficient spatiotemporal inductive biases.

## 5 CONCLUSION

We proposed SplineGS, which utilizes a smooth representation in both space and time for the trajectory of each 3D Gaussian blob, to reconstruct dynamic scenes. Thanks to this representation, the proposed method achieves state-of-the-art or at least competitive performance without the need for a regularizer on the trajectories. The proposed spline representation is an explicit representation, in that the NURBS directly represents the trajectories of Gaussian blobs. This is well suited for dynamic scene reconstruction, providing fast convergence. The proposed model only handles deformations in positions and rotations, so it has limitations in reconstructing objects that appear or disappear in scenes, as well as objects with rapidly changing colors due to lighting variations. Incorporating advanced light reflections and accounting for changes in blob density will be explored in future work.

### ACKNOWLEDGMENTS

This work was partly supported by Innovative Human Resource Development for Local Intellectualization program through the Institute of Information & Communications Technology Planning & Evaluation(IITP) grant funded by the Korea government(MSIT) (IITP-2025-RS-2020-II201741, 50%) and partly by the National Research Foundation of Korea(NRF) grant funded by the Korea government(MSIT) (RS-2023-NR076576, 50%)

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

# APPENDIX

## A   EXCLUSION OF SCALE VARIATIONS IN SPLINEGS

SplineGS does not model the scale variations of Gaussian blobs. This can prohibit certain objects, such as jelly- or smoke-like structures. However, for most types of objects, their parts exhibit minimal scaling changes. It is also reported in the literature that most non-rigid deformations are likely rigid transforms at infinitesimal levels (Lee et al., 2016). This perspective is also adequate for Gaussian splatting, considering that the blobs represent some small parts forming the objects in the scene. Therefore, we concentrate on modeling deformations in positions and rotations only. In reality, modeling scale variations tend to induce overfitting in our experience due to the additional degrees of freedom. We believe that scaling variations should be addressed with careful consideration as a separate research focus.

## B   DISCUSSION ON COLOR AND DENSITY MODELING

In real-world scenarios, dynamic scenes involve not only geometric deformations of objects but also variations in color and density. Incorporating a model that accounts for these factors can enhance performance. However, color and density exhibit distinct behaviors that require additional considerations. Color is highly sensitive to changes in viewpoint and lighting conditions, requiring a model that can capture high-frequency variation. Likewise, density changes, such as the appearance or disappearance of objects, also require the capacity to represent high-frequency variation. Furthermore, the existing densification strategy in 3D-GS is primarily designed for static scenes and does not explicitly handle temporal variations. Without careful consideration of these factors, modeling color and density variations can unintentionally interfere with the representation of physically plausible trajectories. Accordingly, addressing color and opacity variations require additional modelings capable of handling high-frequency changes. Representations such as higher-degree NURBS, Fourier series, or wavelets may offer viable solutions, and further exploration in this direction is left as future work.

## C   MORE RESULTS

Table 5 presents quantitative comparisons of PSNR and SSIM across all scenes in the D-NeRF dataset. In each scene, the proposed method achieves highly competitive results. Table 6 presents quantitative comparisons of LPIPS across all scenes in the D-NeRF dataset. Here, we can confirm that SplineGS also achieves the best LPIPS performance in most cases. Figure 5 shows a scatter plot comparing the number of blobs, FPS, training time, and PSNR of various methods for the D-NeRF dataset. Here, SplineGS has a high average FPS even though its average number of Gaussian blobs is relatively large compared to those of the other methods. This confirms that the core design of the proposed method is indeed quite efficient, and it still achieves high performance. As illustrated in Figure 6, our method produces more accurate renderings than existing approaches. Similarly, Table 7 shows the quantitative comparisons of PSNR and SSIM across five scenes in the Neu3D dataset. Notably, our model achieves the best average performance. Figure 7 provides the rendering results for every 75 frames. Table 8 presents a quantitative comparison of PSNR for three vrig scenes in the HyperNeRF (Park et al., 2021b) dataset. Similar to Neu3D, the HyperNeRF dataset provides real-world scenes. Here, SplineGS achieves performance comparable to other methods. A critical problem with the HyperNeRF dataset is that the provided camera pose information is not accurate. An additional camera pose optimization is required to handle this problem, which is another set of problems left as future work. Table 9 shows the per-scene attributes of SplineGS, such as training time, FPS, and number of blobs, for the D-NeRF and Neu3D datasets.

Table 5: **Quantitative comparison for the D-NeRF dataset.** We compared our method to existing methods on 800×800 resolution test images. Here, "40K" and "80K" in the parentheses indicate the numbers of training iterations. The asterisks (*) indicate that the results were adopted from the 4D-GS paper (Wu et al., 2024), while the daggers (†) indicate that those were from the original papers. All the other results were reproduced in our experiments. The average PSNR and SSIM are measured across all scenes, with some cells highlighted to represent the `best`, `second best`, and `third best`.

| Method | Trex PSNR↑ | Trex SSIM↑ | Jumping Jacks PSNR↑ | Jumping Jacks SSIM↑ | Hell Warrior PSNR↑ | Hell Warrior SSIM↑ |
|---|---|---|---|---|---|---|
| K-Planes[*] (Fridovich-Keil et al., 2023) | 30.43 | 0.9737 | 31.11 | 0.9708 | 24.58 | 0.9520 |
| HexPlane[*] (Cao & Johnson, 2023) | 30.67 | 0.9749 | 31.31 | 0.9729 | 24.55 | 0.9443 |
| TiNeuVox[*] (Fang et al., 2022) | 31.25 | 0.9666 | 33.49 | 0.9771 | 27.10 | 0.9638 |
| D-3DGS[†] (Yang et al., 2024b) | 38.10 | 0.9933 | 37.72 | 0.9897 | 41.54 | 0.9873 |
| 4D-GS[*] (Wu et al., 2024) | 34.23 | 0.9850 | 35.42 | 0.9857 | 28.71 | 0.9733 |
| 4DGS (Yang et al., 2024a) | 30.38 | 0.9743 | 32.10 | 0.9639 | 34.32 | 0.9536 |
| CoGS[†] (Yu et al., 2024) | 37.25 | 0.9923 | 37.48 | 0.9891 | 40.43 | 0.9812 |
| NPGs[†] (Das et al., 2024) | 32.10 | 0.9818 | 33.97 | 0.9828 | 38.68 | 0.9780 |
| CompDynGS[†] (Katsumata et al., 2024) | 28.17 | 0.9740 | 32.93 | 0.9840 | 35.36 | 0.9650 |
| SC-GS (Huang et al., 2024) | 39.61 | 0.9981 | 39.43 | 0.9964 | 42.20 | 0.9925 |
| SplineGS (40K) | 37.96 | 0.9925 | 38.29 | 0.9906 | 42.26 | 0.9890 |
| SplineGS (80K) | 38.77 | 0.9931 | 38.66 | 0.9908 | 42.52 | 0.9893 |

| Method | Stand up PSNR↑ | Stand up SSIM↑ | Bouncing Balls PSNR↑ | Bouncing Balls SSIM↑ | Mutant PSNR↑ | Mutant SSIM↑ |
|---|---|---|---|---|---|---|
| K-Planes[*] (Fridovich-Keil et al., 2023) | 33.10 | 0.9793 | 40.05 | 0.9934 | 32.50 | 0.9713 |
| HexPlane[*] (Cao & Johnson, 2023) | 34.40 | 0.9839 | 39.86 | 0.9915 | 33.67 | 0.9802 |
| TiNeuVox[*] (Fang et al., 2022) | 34.61 | 0.9797 | 40.23 | 0.9926 | 30.87 | 0.9607 |
| D-3DGS[†] (Yang et al., 2024b) | 44.62 | 0.9951 | 41.01 | 0.9953 | 42.63 | 0.9951 |
| 4D-GS[*] (Wu et al., 2024) | 38.11 | 0.9898 | 40.62 | 0.9942 | 37.59 | 0.9880 |
| 4DGS (Yang et al., 2024a) | 38.83 | 0.9857 | 33.28 | 0.9842 | 37.43 | 0.9842 |
| CoGS[†] (Yu et al., 2024) | 43.35 | 0.9929 | 40.98 | 0.9958 | 42.14 | 0.9937 |
| NPGs[†] (Das et al., 2024) | 38.20 | 0.9889 | - | - | 36.02 | 0.9840 |
| CompDynGS[†] (Katsumata et al., 2024) | 40.21 | 0.9940 | 33.29 | 0.9840 | 38.04 | 0.9940 |
| SC-GS (Huang et al., 2024) | 46.47 | 0.9989 | 41.39 | 0.9960 | 43.42 | 0.9989 |
| SplineGS (40K) | 45.50 | 0.9940 | 41.71 | 0.9956 | 43.29 | 0.9960 |
| SplineGS (80K) | 46.00 | 0.9944 | 41.81 | 0.9956 | 43.73 | 0.9962 |

| Method | Hook PSNR↑ | Hook SSIM↑ | Lego PSNR↑ | Lego SSIM↑ | Mean PSNR↑ | Mean SSIM↑ |
|---|---|---|---|---|---|---|
| K-Planes[*] (Fridovich-Keil et al., 2023) | 28.12 | 0.9489 | 25.49 | 0.9483 | 30.67 | 0.9672 |
| HexPlane[*] (Cao & Johnson, 2023) | 28.63 | 0.9572 | 25.10 | 0.9388 | 31.02 | 0.9680 |
| TiNeuVox[*] (Fang et al., 2022) | 28.63 | 0.9433 | 24.65 | 0.9063 | 31.35 | 0.9613 |
| D-3DGS[†] (Yang et al., 2024b) | 37.42 | 0.9867 | 24.94 | 0.9432 | 38.50 | 0.9857 |
| 4D-GS* (Wu et al., 2024) | 32.73 | 0.9760 | 25.03 | 0.9378 | 34.06 | 0.9787 |
| 4DGS (Yang et al., 2024a) | 31.92 | 0.9546 | 24.70 | 0.9189 | 32.87 | 0.9649 |
| CoGS[†] (Yu et al., 2024) | 36.43 | 0.9838 | 25.16 | 0.9451 | 37.90 | 0.9842 |
| NPGs[†] (Das et al., 2024) | 33.39 | 0.9735 | 24.63 | 0.9312 | - | - |
| CompDynGS[†] (Katsumata et al., 2024) | 33.43 | 0.9810 | 24.26 | 0.9400 | 33.21 | 0.9770 |
| SC-GS (Huang et al., 2024) | 38.88 | 0.9956 | 24.92 | 0.9485 | 39.53 | 0.9906 |
| SplineGS (40K) | 38.61 | 0.9891 | 25.29 | 0.9457 | 39.11 | 0.9866 |
| SplineGS (80K) | 38.71 | 0.9891 | 25.29 | 0.9457 | 39.44 | 0.9868 |

Table 6: **LPIPS comparison for the D-NeRF dataset.** We compared our method to existing methods on $800 \times 800$ resolution images based on LPIPS. The daggers (†) indicate that those were from the original papers.

| Method | Trex | Jumping Jakcs | Hell Warrior |
|---|---|---|---|
| 4D-GS† | 0.0131 | 0.0128 | 0.0369 |
| D-3DGS† | 0.0098 | 0.0126 | 0.0234 |
| SC-GS | 0.0119 | 0.0115 | 0.0280 |
| SplineGS | 0.0047 | 0.0065 | 0.0091 |
| Method | Stand up | Bouncing Balls | Mutant |
| 4D-GS† | 0.0074 | 0.0155 | 0.0167 |
| D-3DGS† | 0.0063 | 0.0093 | 0.0052 |
| SC-GS | 0.0072 | 0.0216 | 0.0069 |
| SplineGS | 0.0027 | 0.0025 | 0.0019 |
| Method | Hook | Lego | Mean |
| 4D-GS† | 0.0272 | 0.0382 | 0.0210 |
| D-3DGS† | 0.0144 | 0.0183 | 0.0124 |
| SC-GS | 0.0139 | 0.0499 | 0.0119 |
| SplineGS | 0.0065 | 0.0298 | 0.0080 |

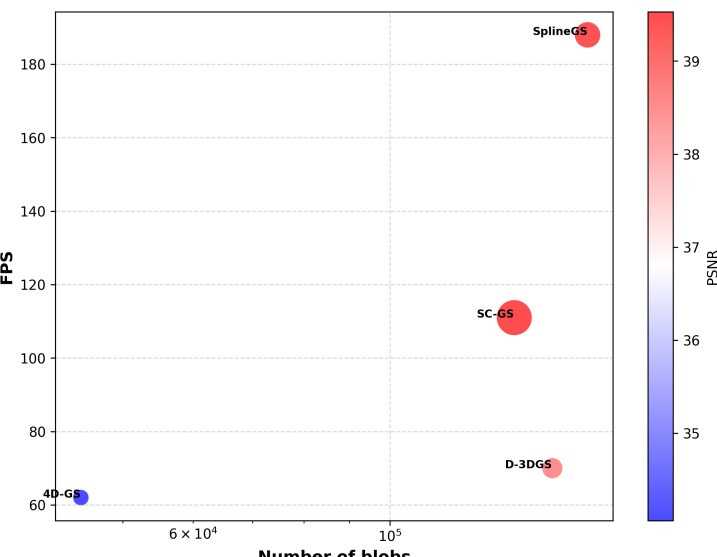

Figure 5: **A scatter plot of various methods based on the number of blobs, FPS, training time, and PSNR.** We compared SplineGS to existing methods on a scatter plot for the D-NeRF dataset. The x-axis represents the average number of blobs, and the y-axis represents the average FPS. The size of each circle indicates the average training time, while the color represents the average PSNR.

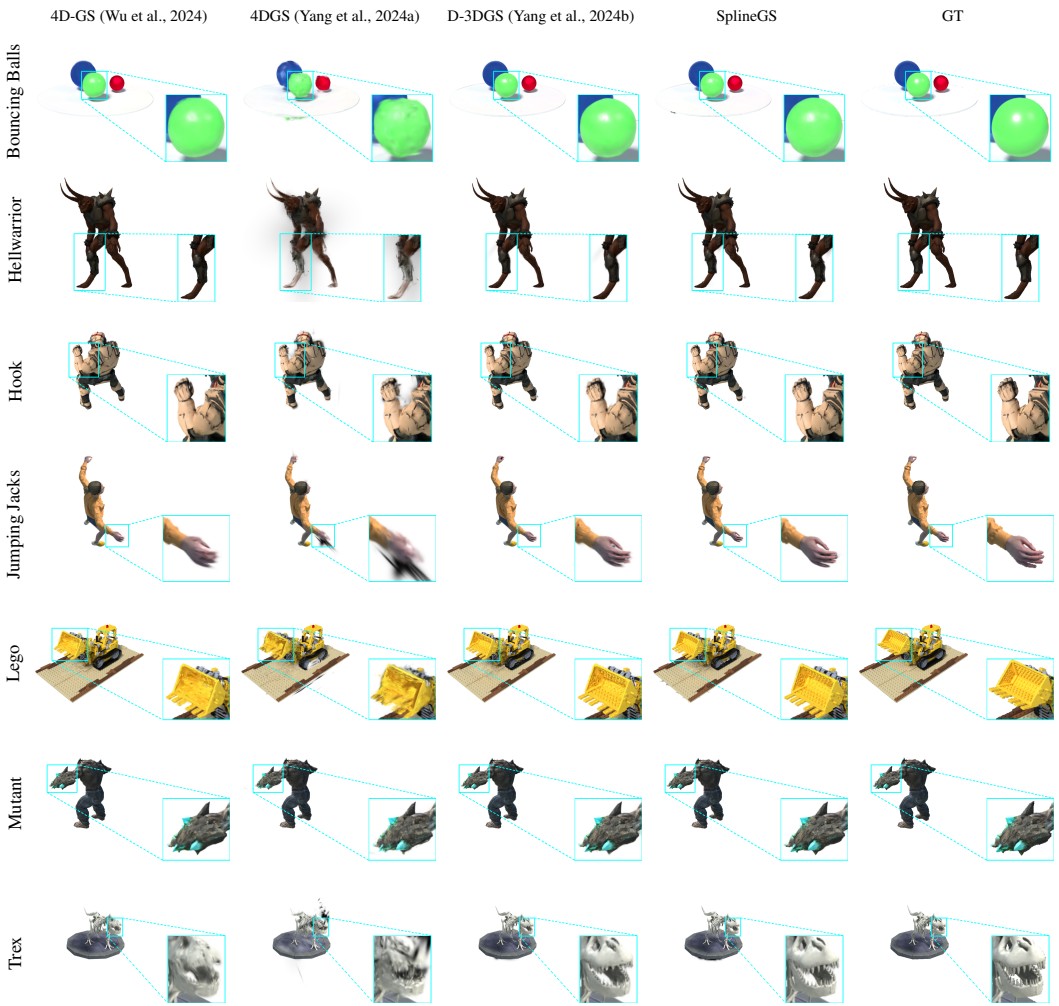

Figure 6: **Qualitative results on the D-NeRF dataset.** We present the rendering results of various methods, including ours, for all scenes in the D-NeRF dataset. The first to third columns show the results of existing methods, while the fourth and fifth columns show those of the proposed method and the ground truth, respectively. Each row shows the results of a different scene.

Table 7: **Quantitative comparison for the Neu3D dataset.** We compared our method with existing methods on 1352×1014 resolution test images. Here, "20K" and "40K" in the parentheses indicate the numbers of training iterations. The conventions for typographical marks and color codes are generally identical to Table 1. We measured the average PSNR and SSIM across all scenes except *flame_salmon*, following the practice of many existing works (Guo et al., 2024; Liu et al., 2024; Lu et al., 2024).

| Method | Coffee Martini PSNR↑ | SSIM↑ | Spinach PSNR↑ | SSIM↑ | Cut Beef PSNR↑ | SSIM↑ |
|---|---|---|---|---|---|---|
| NeRFPlayer[*] (Song et al., 2023) | 32.05 | 0.9380 | 32.06 | 0.9300 | 31.83 | 0.9280 |
| K-Planes[*] (Fridovich-Keil et al., 2023) | 29.99 | 0.9530 | 32.60 | 0.9660 | 31.82 | 0.9660 |
| HexPlane[*] (Cao & Johnson, 2023) | - | - | 31.86 | 0.9830 | 32.71 | 0.9850 |
| 4D-GS[*] (Wu et al., 2024) | 27.34 | 0.9050 | 32.46 | 0.9490 | 32.90 | 0.9570 |
| 4DGS[†] (Yang et al., 2024a) | 28.33 | - | 32.93 | - | 33.85 | 0.9800 |
| STG[†] (Li et al., 2024) | 28.61 | 0.9585 | 33.18 | 0.9785 | 33.52 | 0.9795 |
| SplineGS (20K) | 29.16 | 0.9171 | 33.04 | 0.9538 | 33.42 | 0.9541 |
| SplineGS (40K) | 29.26 | 0.9189 | 33.09 | 0.9542 | 33.49 | 0.9544 |

| Method | Sear Steak PSNR↑ | SSIM↑ | Flame Steak PSNR↑ | SSIM↑ | Mean PSNR↑ | SSIM↑ |
|---|---|---|---|---|---|---|
| NeRFPlayer[*] (Song et al., 2023) | 32.31 | 0.9400 | 27.36 | 0.8670 | 32.12 | 0.9206 |
| K-Planes[*] (Fridovich-Keil et al., 2023) | 32.52 | 0.9740 | 32.39 | 0.9700 | 31.86 | 0.9658 |
| HexPlane[*] (Cao & Johnson, 2023) | 32.09 | 0.9860 | 31.92 | 0.9880 | - | - |
| 4D-GS[*] (Wu et al., 2024) | 32.49 | 0.9570 | 32.51 | 0.9540 | 31.54 | 0.9444 |
| 4DGS[†] (Yang et al., 2024a) | 33.51 | - | 34.03 | - | 32.53 | - |
| STG[†] (Li et al., 2024) | 33.89 | 0.9826 | 33.64 | 0.9824 | 32.57 | 0.9740 |
| SplineGS (20K) | 33.75 | 0.9621 | 33.23 | 0.9578 | 32.52 | 0.9484 |
| SplineGS (40K) | 33.82 | 0.9624 | 33.33 | 0.9582 | 32.60 | 0.9496 |

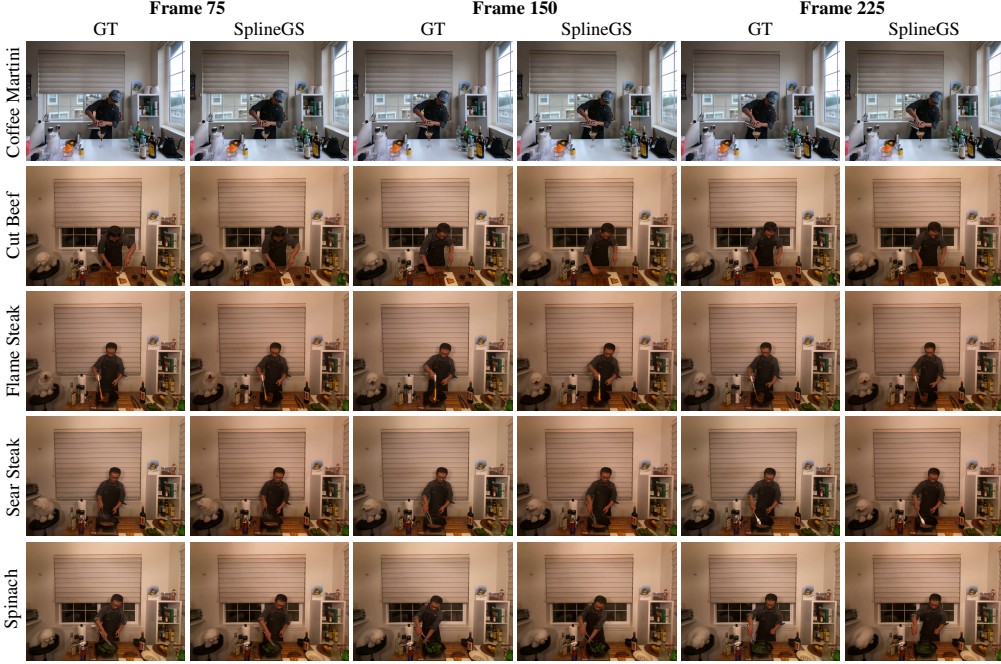

Figure 7: **Qualitative results on the Neu3D dataset.** We show the qualitative results of SplineGS on the Neu3D datasets except for *flame_salmon*. We provide the ground truth and the results of SplineGS for the 75th, 150th, and 225th frames. Each row shows the results of a different scene.

Table 8: **Quantitative comparison for the HyperNeRF dataset.** We compared our method to existing methods on half-resolution test images and measured PSNR for three vrig scenes. The daggers (†) indicate that those were from the original papers.

| Method | 3D Printer | Chicken | Broom |
|---|---|---|---|
| D-3DGS (Yang et al., 2024b) | 20.28 | 22.76 | 20.47 |
| 4D-GS† (Wu et al., 2024) | 22.10 | 28.70 | 22.00 |
| SplineGS | 21.97 | 29.13 | 20.56 |

Table 9: **Per-scene attributes of SplineGS for the D-NeRF and Neu3D datasets.** We present the training time, FPS, and number of blobs for each scene. The number of representative trajectories $L$ was $L = 64$ for *mutant* and *bouncing balls*, while that was $L = 128$ for the other scenes.

| D-NeRF | | | | Neu3D | | | |
|---|---|---|---|---|---|---|---|
| Scene | Time (min) | FPS | #Blobs | Scene | Time (min) | FPS | #Blobs |
| Trex | 41 | 133 | 250K | Coffee Martini | 67 | 62 | 386K |
| Jumping Jack | 33 | 197 | 126K | Spinach | 53 | 81 | 140K |
| Hell Warrior | 28 | 263 | 52K | Cut Beef | 52 | 79 | 167K |
| Stand Up | 31 | 227 | 98K | Sear Steak | 52 | 84 | 138K |
| Bouncing balls | 38 | 176 | 167K | Flame Steak | 50 | 76 | 143K |
| Mutant | 34 | 192 | 165K | | | | |
| Hook | 33 | 186 | 161K | | | | |
| Lego | 42 | 133 | 320K | | | | |

