# OpenReview forum: "SplineGS: Learning Smooth Trajectories in Gaussian Splatting for Dynamic Scene Reconstruction"
_ICLR.cc/2025/Conference — ICLR 2025 Poster_

### Official Review · Reviewer_J4YF · 2024-10-28

**Soundness:** 2
**Presentation:** 2
**Contribution:** 2
**Rating:** 6
**Confidence:** 4

**Summary:**

This paper proposes SplineGS, a model that enables Gaussian Splatting to handle trajectories of Gaussian smoothly. SplineGS first predict position and rotation using an MLP over time and employ non-uniform rational B-splines.

**Strengths:**

1. SplineGS enables smooth representation of dynamic scenes without regulerizations.

2. SplineGS provides competitive performance compared to the existing methods.

**Weaknesses:**

1. Lacks significant advantages over other models. For example, SC-GS shows better rendering quality on the D-NeRF dataset, and on the Neu3D dataset, SplineGS’ PSNR is only 0.03 dB higher than STG. It seems like SplineGS' main contribution is creating smooth trajectories for Gaussians, but only the visualization in Figure 1 supports this. I suggest adding visualizations that shows the Gaussians being learned in a smooth trajectory on various scenes.

2. In line 85, the B-spline functions as a smooth regularizer, but according to line 231, rendering requires $p + M + 1$ additional points. Using the smooth regularizer should allow rendering with just one point, which seems inefficient. Therefore, a comparison of the trade-off between rendering speed and rendering quality is necessary, especially in contrast to using the regularizer.

3. The appearance or disappearance of objects cannot be handled, as mentioned in the conclusion. However, in real-world scenes like Neu3D, I believe that such cases also exist, but there is a lack of analysis on this aspect. In Fig. 6, the flame streak scene in Neural 3D Video shows flames appearing and disappearing, suggesting that the method may handle disappearing objects. However, it remains unclear how SplineGS manages this. I recommend an analysis, such as visualizing the trajectories of the Gaussians.

4. SplineGS performs video view synthesis, but no video results are provided.

**Questions:**

1. I believe that for the Flame Salmon scene, experiments can be conducted using only the first 300 frames, as done in 4DGS, STG, and Gaussian-Flow. Could you provide results of the scene?

2. Are there trajectory results on the Neu3D dataset?

3. Can you provide LPIPS results?

---

> ### Author Response · Authors · 2024-11-20
>
> ### W1. Small performance gap / visualizations.
> SplineGS demonstrates comparable performance with a clear advantage in training time. This suggests that the proposed representation is quite effective in modeling natural deformations, enabling efficient training. Moreover, some models in Table 2 (4DGS and STG) incorporate additional models for color and density variations, unlike SplineGS where only positions and rotations are considered. SplineGS still beats these methods, which also confirms the effectiveness of the proposed trajectory representation.
>
> Here, we provide a video comparing the qualitative results of SplineGS and two of the existing methods, which can also be found in the new supplementary material. Please refer to the answer to W4 for the visualization video.
>
> ### W2. Rendering efficiency based on B-spline.
> It is true that the calculation of basis functions requires $p+M+1$ knots, but this calculation is needed only once BEFORE training begins because the basis functions do not change during training. During training, a single rendering requires only $p+1$ nearby control points, i.e., defining the entire trajectory requires $M$ control points but only $p+1$ points are required to calculate the value at a specific time instance. Moreover, this calculation is only needed for representative trajectories, of which the number $L$ is much fewer than the number of blobs $N$. Accordingly, $M$ does not affect the rendering speed. $p$ is usually set to three in general usages of B-splines (using 3rd-degree polynomials), which we have also followed in SplineGS.
>
> Performance-wise, increasing $M$ does not always result in better performance, because too large $M$ can cause overfitting in our experience. Hence, it is not really appropriate to discuss a tradeoff in this regard. Instead, here we provide a scatter plot [(anonymous link)](https://anonymous.4open.science/r/SplineGS-0AF1/Figure5.png) illustrating FPS, the number of blobs, training time, and PSNR of various methods, which is also added to Figure 5 in the appendix. In the plot, we can confirm that SplineGS has a high average FPS even though it uses large numbers of Gaussian blobs compared to the other methods. This demonstrates the inherent efficiency of the proposed representation. In the plot, we can also confirm that SplineGS provides high performance with moderate training time compared to the existing methods.
>
> ### W3. The appearance or disappearance of objects.
> We thank the reviewer for this insight. SplineGS indeed does not include modeling for density changes. For the flame steak scene, it seems that Gaussians with low opacity temporarily overlap and dissipate in SplineGS, creating the effect of flames. The main focus of this paper is to provide an effective representation of trajectories in Gaussian splatting, so modeling the changes in color or density is left as future work. Nevertheless, SplineGS still beats the other methods for Neu3D in Table 2, which demonstrates the effectiveness of the proposed trajectory representation. A new supplementary video visualizing the trajectories has been added, as described in the next answer.
>
> ### W4. Providing visualization videos.
> Here, we provide a video [(anonymous link)](https://anonymous.4open.science/r/SplineGS-0AF1/SplineGS_video.mp4) comparing the qualitative results of SplineGS and two of the existing methods, which can also be found in the new supplementary material. The video contains the visualizations for the D-NeRF and Neu3D datasets, which demonstrates that SplineGS can provide high-quality results with natural, realistic trajectories based on NURBS. Notably, SplineGS produces smooth trajectories even when increasing the rendering frame rate.
>
> ### Q1. Results of Flame Salmon with the first 300 frames.
> SplineGS achieves a PSNR of 29.9 and an SSIM of 0.9237 on the first 300 frames of the Flame Salmon scene. In comparison, STG achieves a PSNR of 29.17 when trained with a single model on 300 frames and 29.48 when trained with one model per 50 frames.
>
> ### Q2. Trajectory results on Neu3D.
> As described in the answer to W4, we have newly provided a supplementary video. Here, the trajectory results for Neu3D are also provided.

---

> ### Author Response · Authors · 2024-11-20
>
> ### Q3. LPIPS results.
> Here, we provide the LPIPS results as requested (which have also been added to Table 7 in the appendix). SplineGS achieves the best performance in most cases.
>
> | Method   | Trex   | Jumping Jacks | Hell Warrior | Stand up | Bouncing Balls | Mutant | Hook   | Lego   | Mean   |
> |----------|--------|---------------|--------------|----------|----------------|--------|--------|--------|--------|
> | 4D-GS   | 0.0131 | 0.0128        | 0.0369       | 0.0074   | 0.0155         | 0.0167 | 0.0272 | 0.0382 | 0.0210 |
> | D-3DGS  | 0.0098 | 0.0126        | 0.0234       | 0.0063   | 0.0093         | 0.0052 | 0.0144 | 0.0183 | 0.0124 |
> | SC-GS    | 0.0119 | 0.0115        | 0.0280       | 0.0072   | 0.0216         | 0.0069 | 0.0139 | 0.0499 | 0.0119 |
> | SplineGS | 0.0047 | 0.0065        | 0.0091       | 0.0027   | 0.0025         | 0.0019 | 0.0065 | 0.0298 | 0.0080 |

---

> > ### Comment · Reviewer_J4YF · 2024-11-23
> >
> > I appreciate the author's detailed explanations and the materials provided. However, there remains concern due to limited novelty.
> >
> > The author claims that the rendering speed is faster than 4DGS inherits from the proposed representation, but this seems to be due to the advantages of "fully fused MLP" from the "tiny-cuda-nn" implementation [1] (it shows more than $5\times$ the speed-up). The introduction of the hash grid representation is also one of the author's contributions, but NeRFPlayer already introduced the hash grid representation in dynamic scene reconstruction, and the overall pipeline that deforms the space was introduced in 4D-GS. According to Table 4, in the case of real-world scenes, the performance change due to the hash grid representation is greater than that of NURBS, and I expect the PSNR to be similar or lower than that of 4D-GS if the hash grid representation is not used. Therefore, SplineGS is highly dependent on existing implementations, and NURBS is effective only for certain scenes (synthetic cases), so its contribution does not seem significant.
> >
> > [1] Thomas Müller et.al., "Real-time neural radiance caching for path tracing", ACM Trans. Graph. 40, 2021

---

> > > ### Author Response · Authors · 2024-11-24
> > >
> > > J4YF also argues that the overall deformation pipeline is from 4D-GS, however, this point is shared across most works in dynamic Gaussian splatting literature, except 4DGS. Accordingly, this says nothing about the proposed method. Overall, we cannot really agree with the `the basic framework is from another work' tone in J4YF's arguments. Changes implemented within the frameworks are what make a real difference. In the case of SplineGS, this is the efficient implicit regularization provided by the proposed trajectory representation, which is the reason for achieving SOTA performance without any heavy burdens.

---

> > > > ### Comment · Reviewer_J4YF · 2024-11-25
> > > >
> > > > As you mentioned, I am not claiming that the hash grid representation is the contribution of this paper, and this is not to deny the contribution of the framework that makes smooth trajectory of Gaussian based on spline. I think the most important claim of this paper is the smooth representation by NURBS, but this paper does not sufficiently demonstrate the importance of smooth trajectory. This is because the smoothing of the trajectory is only applied to the limited baseline, making it hard to compare with other methods and therefore difficult to see its own advantages.
> > > >
> > > > The basic framework I meant is a structure that represents a dynamic scene with Gaussian Splatting by predicting the transformation of position and rotation from spatial position and time (4D-GS attempted to combine Gaussian Splatting and MLP in dynamic scene reconstruction). The method proposed in this paper is to predict the transformation of position and rotation multiple times over time and then generate a smooth trajectory with a spline. Therefore, this work seems to be an extension of 4D-GS.

---

> > > > > ### Author Response · Authors · 2024-11-28
> > > > >
> > > > > Dear Reviewer,
> > > > >
> > > > > We have left more responses regarding your last reply. We hope that the above responses fully address your remaining concerns. We would greatly appreciate your feedback during the extended discussion period.

---

> ### Author Response · Authors · 2024-11-24
>
> We believe Reviewer J4YF has misread our argument regarding the contribution. First of all, using the hash grid representation is definitely not our main contribution, of course, and what we are arguing in the corresponding statement (at the end of Introduction) is that letting the hash grid + MLP framework `output the weights for linear combination' is our contribution. This allows us to enforce spatial smoothness implicitly with little additional computational burden.
>
> It is true that the proposed method is built upon the efficient implementation of the fully fused MLP, however, our contribution regarding the rendering speed, again, lies in the design that comes AFTER the MLP. SplineGS does not require $O(n\log n)$ comparisons to achieve spatial smoothness, and its computational complexity is proportional to the number of representative trajectories $L$, where $L$ is much smaller than the number of blobs $n$ ($L \ll n$). In other words, the proposed trajectory representation adds little to the overall processing time. The following table shows the comparison between SplineGS and that without the proposed representation (no NURBS and no representative trajectories).
>
> | Method    | Bouncing Balls (FPS) | Coffee Martini (FPS) |
> |-----------|----------------|----------------|
> | w/o NURBS | 201            | 58             |
> | Proposed  | 176            | 62             |
>
> Comparing these, we can confirm that the proposed representation takes only a small fraction of the overall processing time, meaning that it can be applied to any other framework without much burden. Interestingly, the rendering speed even drops for Coffee Martini without the representation. In this case, many spurious blobs are produced due to the lack of proper inductive bias, increasing the processing time. This also highlights the importance of a proper trajectory representation.
>
> Similarly, J4YF's argument regarding the contribution in terms of performance (i.e., `it is not significant compared to the (bare-)hash-grid framework') is seriously misleading. First of all, it is not true: In Table 4, for Neu3D (the real dataset), the use of the proposed representation has non-negligible PSNR gaps (0.5 to 1.8), i.e., using the hash grid framework alone does not perform as well as SOTA. Here are the full ablation study results for Neu3D, which show up to 2.0 PSNR gaps when the proposed representation is not used:
>
> | Method      | Coffee Martini (PSNR) | Coffee Martini (SSIM) | Spinach (PSNR) | Spinach (SSIM) | Cut Beef (PSNR) | Cut Beef (SSIM) | Sear Steak (PSNR) | Sear Steak (SSIM) | Flame Steak (PSNR) | Flame Steak (SSIM) |
> |-------------|---------------|---------------|----------------|----------------|-----------------|-----------------|-------------------|-------------------|---------------------|---------------------|
> | w/o hash    | 27.98         | 0.9041        | 30.04          | 0.9385         | 28.47           | 0.9334          | 30.87            | 0.9473            | 29.46              | 0.942              |
> | w/o MLP     | 28.80         | 0.9160        | 32.28          | 0.9507         | 32.60           | 0.9523          | 33.19            | 0.9605            | 31.75              | 0.9536             |
> | w/o NURBS   | 28.85         | 0.9181        | 32.08          | 0.9498         | 31.45           | 0.9460          | 32.08            | 0.9559            | 31.25              | 0.9520             |
> | Proposed    | 29.26         | 0.9189        | 33.09          | 0.9542         | 33.49           | 0.9544          | 33.82            | 0.9624            | 33.33              | 0.9582             |
>
> In fact, many works have also used the hash grid framework, but they usually rely solely on neural networks for deformation estimation, showing limitations in performance. This highlights the need for a proper representation of trajectories. Meanwhile, the poor performance without the hash grid does NOT mean what J4YF argued (i.e., `only the hash grid is dominant'), either: Here, it has been replaced with a quite primitive baseline (simple positional encodings of the mean positions), hence, it is obvious that performance drops a lot; which proves nothing about the importance of the proposed representation. The really important question here is; the hash-grid framework alone is not good enough for SOTA, so what should we add to achieve SOTA performance? This is what the proposed trajectory representation is just for.

---

> ### Author Response · Authors · 2024-11-26
>
> ### ● Advantages of smooth trajectory.
> We have already shown the advantages of smooth trajectory representation multiple times with concrete evidence; however, it seems that the significance of these points has not yet been noticed by J4YF. We have shown in the above table (the full ablation study results on Neu3D) that applying NURBS improves performance up to 2.0 PSNR. This is a very clear result, demonstrating the advantage of using NURBS.
>
> Moreover, we have also shown the advantage of the proposed representation in the supplementary video as well. Notably, SplineGS produces smooth trajectories even when increasing the rendering frame rate. The video further illustrates the capability of NURBS to maintain smoothness while capturing fine details, even in the more complex Neu3D dataset. Overall, the video provides visual evidence to support our claim.
>
> It is really hard to grasp what J4YF's criticism points on. What does it mean that the proposed representation is 'only applied to limited baseline and therefore hard to compare with other methods'? We cannot really agree on this point because we have compared the proposed method with the most recent methods on all the popular benchmark datasets, accompanied by a detailed ablation study. Please elaborate on this if J4YF still thinks the other way.
>
> ### ● Deforming canonical space.
> J4YF keeps saying that SplineGS seems to be an extension of 4D-GS, but as we already pointed out, 4D-GS is not the only method that deforms the canonical space based on a neural network, and actually, this is a very common approach in dynamic 3D reconstruction. It actually dates back to D-NeRF (Pumarola et al., 2021), where deformation is predicted by an MLP and added to the canonical space as follows:
>
> $$
> x(t) = x_{c} + \Psi_t (x_c, t)
> $$
>
> where $x_c$ represents sampled points along a ray. Similarly, many NeRF-based methods, such as NeRFPlayer (Song et al., 2023), adopt a deforming canonical space approach. In the 3D-GS framework, D-3DGS (Yang et al., 2024b) is actually the first work applying this type of canonical space-based deformation modeling: It predicts the changes of position, scaling, and rotation from spatial position and time:
>
> $$
> (\delta x, \delta r, \delta s) = \mathcal{F}_\theta (\gamma (\mathrm{sg}(x)), \gamma (t))
> $$
>
> This type of deformation modeling is largely shared across many dynamic 3D-GS methods, including SC-GS  (Huang et al., 2024), 3D geometry-aware deformable Gaussian splatting (Lu et al., 2024), CoGS (Yu et al., 2024), and 4D-GS (Wu et al., 2024). Accordingly, it is a very narrow view that the canonical space-based approach is only limited to 4D-GS. It is even more misleading to say that SplineGS is an extension of 4D-GS because of this. As explained above, the canonical space-based deformation framework is merely a basic framework that has been shared across many works in this field for many years. Accordingly, we cannot really agree on this point. Please elaborate on this if J4YF still thinks the other way.

---

> ### Author Response · Authors · 2024-11-28
>
> We believe that Reviewer J4YF's concerns come from a misunderstanding of the proposed method. The summary provided by J4YF, ``"SplineGS first predicts position and rotation using an MLP over time and employs non-uniform rational B-splines."`` significantly mismatches our method. In our pipeline, the MLP outputs the weights for the linear combination of representative trajectories and it does not receive any time information. The representative trajectories are defined by NURBS, from which the deformations of positions and rotations are computed, and they are defined independently with MLP. The deadline for the discussion period is approaching, and we are waiting for J4YF's participation to continue the discussion.

---

> ### Comment · Reviewer_J4YF · 2024-11-28
>
> I appreciate the author's response. The misunderstanding of the model's architecture seems to have come from the fact that t in Figure 2 does not seem to be the position at time t obtained from the spline, but rather the value fed to the model. After seeing this, I understood the description in 3.2 as the creation of control points according to t and the behavior of linear combinations. I recommend changing the arrow of t in Figure 2 to a dash line or something similar (this should be enough to indicate the point at time t in the spline). This really leads to a misunderstanding.
>
> This misunderstanding has been resolved, which has also solved other questions, but I still have a few suggestions. I believe that this methodology is general, but it only seems to be valid for hash grid representation, so it feels like this model relies on hash grid. In particular, in the case of a sufficiently small scene like a synthetic scene, there is no performance gap because MLP can handle it sufficiently even without hash, but in the real world, I suspect that this gap is larger because the parameters of MLP do not increase as much as the parameters of hash grid are reduced. This makes the experiment seem unfair. Experiments with other representations such as voxel grid or TensoRF would also be good for demonstrating the generality of the model. Also, emphasizing that precomputation of splines created by deformation fields is possible would be a good explanation of the benefits of FPS. Skipping the calculation of MLP over time when rendering is a great advantage. The remaining questions were sufficiently addressed by the answers from other reviewers. I have therefore raised my score.

---

> ### Author Response · Authors · 2024-11-30
>
> We are pleased to hear that the reviewer's concern has been addressed. As the reviewer suggested, we will revise Figure 2 in the camera-ready submission. As J4YF pointed out, the NURBS-based trajectory is defined independently of the neural network, allowing the use of voxel grids or TensorRF as alternatives to the hash grid. However, this would shift the focus to a comparison of existing spatial scene representations rather than validating the new temporal smooth representation. Nonetheless, we appreciate your suggestion, and we will consider this point in future work.

---

### Official Review · Reviewer_x185 · 2024-10-31

**Soundness:** 2
**Presentation:** 3
**Contribution:** 2
**Rating:** 6
**Confidence:** 5

**Summary:**

This paper proposes SplineGS, which reconstructs dynamic scenes using the non-uniform rational B-splines (NURBS).
SplineGS can learn temporally smooth trajectories based on NURBS,  and learn weights of the combinations by a multi-resolution hash table and an MLP.
SplineGS leads to high-fidelity rendering and fast training times.

**Strengths:**

- SplineGS can obtain smooth trajectories utilizing NURBS.
- The paper is well written, and the idea is easy to understand.

**Weaknesses:**

- The novelty is limited.
   1. Although SplineGS does not need any regularizers for trajectories, NURBS also can be treated as a regularizer. More importantly, this paper does not demonstrate the usefulness of the learned smooth trajectories.
   2. To obtain the weights, SplineGS employs a multi-resolution hash table and an MLP, which can be treated as learning the weights of LBS through the neural network. Compared to other methods of obtaining LBS weights (such as Gaussian kernel RBF), what are the advantages of using neural networks?
  3. In my opinion, the shorter training time and higher FPS are more due to L1 regularization of the opacity value in Eq. (15), which reduces the number of Gaussians.
- Although this paper achievies state-of-the-art or at least competitive performance, hower the performance is not consistent. For example, compared with 4DGS (Yang et al., 2024a), SplineGS achieves higher PSNR and FPS on the D-NeRF dataset, while achieving similar PSNR and lower FPS on the Neu3D dataset. Experiments on more real datasets are needed to evaluate the effectiveness of SplineGS.

**Questions:**

See `Weaknesses`. More,
1. The number of Gaussians per scene, the number of representative trajectories, the training time, and the FPS of SplineGS for the D-NeRF dataset and Neu3D dataset should be provided.
2. `LPIPS` metric also should be reported. As shown in D-3DGS, results on the “Lego” scene of the D-NeRF dataset should not be reported, nor should performance on the validation set be reported.
3. I would like to know the performance of replacing the multi-resolution hash table and MLP with the Gaussian kernel RBF adopted by SC-GS.
4. It would be better to provide some videos to intuitively demonstrate the performance of SplineGS.
5. Are there any failure cases? For example, how SplineGS performs on the “vrig-broom” scene on the Hyper-NeRF dataset.
6. Laking the results of 4DGS on the terms of `SSIM` and `Training time` for the Neu3D dataset.

---

> ### Author Response · Authors · 2024-11-20
>
> ### W0. Novelty.
> We strongly disagree with the reviewer regarding the weak-novelty argument. The main benefit SplineGS offers is its clear advantage in training time, which is attributed to many novel ideas in the proposed trajectory representation. We believe that the reviewer has many small misconceptions regarding the details of the proposed method, which has led to this argument in our opinion. We will rebut this point in the following three answers, often with concrete evidence.
>
> ### W1. NURBS as an implicit regularizer / the advantages of NURBS.
> We agree that NURBS itself acts as an implicit regularizer. The no-regularizer argument in the paper was mainly regarding computational complexity introduced by an explicit regularizer, which usually requires either $O(n)$ or $O(n\log n)$ computation. This can be a major issue in processing speed if a large number of blobs are used for high fidelity. On the other hand, NURBS in SplineGS, limitedly used for representative trajectories, requires much less computation while it introduces enough inductive biases for spatiotemporally smooth trajectories. This advantage is especially significant when a large number of blobs are used (refer to the answer to W3). We have added this explanation in Section 1 to clarify this advantage.
>
> The advantages of using NURBS are three-fold: (i) As mentioned above, computational cost can be saved while smoothness regularization is implicitly applied. (ii) NURBS, unlike other basis representations such as Fourier, can model locally concentrated deformations since calculating deformation at a specific time instance only requires $p+1$ nearby control points. (This explanation has been added in Section 2.2.) (iii) The learned smooth trajectories can better represent natural deformations. To demonstrate the last advantage, we have added a new supplementary video (refer to the answer to Q4) comparing the reconstruction results of SplineGS and two existing methods. Here, we can see that the learned trajectories in SplineGS contain less unlikely fluctuations, especially when increasing the rendering frame rate, demonstrating the adequacy of the proposed representation.
>
> ### W2. Advantages of using MLP for the weights of trajectories (spatial smoothness).
> SC-GS uses a kernel regression approach, which requires finding the $K$-nearest neighbors, leading to $O(n \log n)$ complexity where $n$ is the number of blobs. In contrast, SplineGS relies on the capability of MLP to handle this, offering a significant advantage in computational efficiency. Moreover, the similarities between trajectories are not always tied to spatial proximities, i.e., some far-away parts can exhibit similar trajectories in real-world deformations (e.g., different wings of a large fan). The proposed design can be more flexible in handling this kind of deformation, unlike SC-GS. Even though this approach does not directly consider physical information in calculating the weights, the results demonstrate high performance, indicating that it is also highly effective.
>
> ### W3. Impacts of opacity regularization on computational efficiency.
> As mentioned by the reviewer, (15) can contribute to shorter training times and higher FPS; however, our design choices for the trajectory representation play a much more significant role, reducing the fundamental computational complexity. Here, we provide a scatter plot [(anonymous link)](https://anonymous.4open.science/r/SplineGS-0AF1/Figure5.png) illustrating the processing time and the number of blobs for various methods, which has also been added to Figure 5 in the appendix. In the plot, we can confirm that SplineGS has a high average FPS even though it uses large numbers of Gaussian blobs compared to the other methods. This indicates that the improved training time and FPS are not mainly due to the L1 regularization, and the inherent efficiency of the proposed representation plays a significant role here. In the plot, we can also confirm that SplineGS provides high performance with moderate training time compared to the existing methods.

---

> ### Author Response · Authors · 2024-11-20
>
> ### W4. Inconsistencies in Tables 1 and 2 / more real datasets
> Regarding FPS, in reality, it is the other way around. Comparing Tables 1 and 2, 4D-GS (Wu et al., 2024) and SplineGS both show similar tendencies regarding FPS, i.e., they have lower FPS for Neu3D. On the other hand, 4DGS (Yang et al., 2024a) has a relatively higher FPS for Neu3D, and we conjecture the reason to be the characteristics of 4DGS. 4DGS is based on 4D Gaussian blobs, and if the scene contains a large static area (e.g., background), then it is possible that the number of required 4D blobs can be reduced. On the other hand, most other methods explicitly model the trajectories of 3D blobs, and static regions do not provide any advantages in processing speed. In Table 2, STG also has a high FPS and the main reason is its color modeling. STG does not directly define the color of each blob; instead, it first renders the blob features directly to a 2D plane. Then, this rendered 2D feature map goes through an MLP to output color. This requires 9 dimensions for each blob, which is much smaller than the 48 dimensions (3rd-degree spherical harmonics) in SplineGS. While this approach allows faster rendering speeds in novel-view synthesis, it cannot provide a full 3D scene reconstruction, which differs from our goals.
>
> Regarding PSNR, it is true that the performance gap is smaller for Neu3D. The Neu3D dataset, consisting of real data, involves changes in color and opacity. Some models, such as 4DGS (Yang et al., 2024a) and STG, incorporate modeling for these. On the other hand, SplineGS concentrates only on the changes in blob positions and rotations, showing a relatively small performance gap for this dataset compared to that in the D-NeRF dataset. Nevertheless, SplineGS achieves higher performance despite having fewer deformation components, suggesting that the proposed representation is quite effective for trajectory modeling. This discussion was already present in Section 4. Considering color and density changes is a separate research direction that requires careful consideration, which is not the main focus of this paper.
>
> Regarding more experiments on real datasets, we have provided the results on the HyperNeRF dataset in the answer to Q5.
>
> ### Q1. Detailed configuration for each scene.
> Here, we provide the detailed experimental configuration for each scene as requested (also added to Table 5 of the appendix). The number of representative trajectories $L$ was $L=64$ for mutant and bouncing balls, while that was $L=128$ for the other scenes.
>
> | **D-NeRF**        | Time (min) | FPS  | #Blobs  | **Neu3D**         | Time (min) | FPS | #Blobs  |
> |--------------------|------------|-------|---------|-------------------|------------|------|---------|
> | Trex              | 41         | 133   | 250K    | Coffee Martini   | 67         | 62   | 386K    |
> | Jumping Jack      | 33         | 197   | 126K    | Spinach          | 53         | 81   | 140K    |
> | Hell Warrior      | 28         | 263   | 52K     | Cut Beef         | 52         | 79   | 167K    |
> | Stand Up          | 31         | 227   | 98K     | Sear Steak       | 52         | 84   | 138K    |
> | Bouncing balls    | 38         | 176   | 167K    | Flame Steak      | 50         | 76   | 143K    |
> | Mutant            | 34         | 192   | 165K    |                   |            |      |         |
> | Hook              | 33         | 186   | 161K    |                   |            |      |         |
> | Lego              | 42         | 133   | 320K    |                   |            |      |         |
>
> ### Q2. LPIPS / concerns on the Lego scene.
> Here, we provide the LPIPS results as requested (which have also been added to Table 7 in the appendix). SplineGS achieves the best performance in most cases. Regarding the Lego scene, while we agree with the reviewer's concern (discrepancies between train and test), we nevertheless decided to include this because most recent works include this scene in their results.
>
> | Method   | Trex   | Jumping Jacks | Hell Warrior | Stand up | Bouncing Balls | Mutant | Hook   | Lego   | Mean   |
> |----------|--------|---------------|--------------|----------|----------------|--------|--------|--------|--------|
> | 4D-GS   | 0.0131 | 0.0128        | 0.0369       | 0.0074   | 0.0155         | 0.0167 | 0.0272 | 0.0382 | 0.0210 |
> | D-3DGS  | 0.0098 | 0.0126        | 0.0234       | 0.0063   | 0.0093         | 0.0052 | 0.0144 | 0.0183 | 0.0124 |
> | SC-GS    | 0.0119 | 0.0115        | 0.0280       | 0.0072   | 0.0216         | 0.0069 | 0.0139 | 0.0499 | 0.0119 |
> | SplineGS | 0.0047 | 0.0065        | 0.0091       | 0.0027   | 0.0025         | 0.0019 | 0.0065 | 0.0298 | 0.0080 |

---

> ### Author Response · Authors · 2024-11-20
>
> ### Q3. Applying kernel regression to SplineGS.
> Adopting kernel regression in SplineGS is not possible, because the representative trajectories, the subjects of linear combinations in SplineGS, do not have any particular spatial locations. They simply represent modes of deformations without the spatial bias term (i.e., they represent $\Delta X$, which is why their linear combinations must be added to the static mean positions of blobs). Kernel regression is possible in SC-GS only because the trajectories of anchor points are used. We believe this misconception has happened because of a misleading phrase in Section 3.3: ``... and instead, learn the proximities implicitly by the weights produced from the hash table and MLP.`` We have revised this to ``... and instead learn the weights implicitly by the hash table and MLP.`` Even though incorporating kernel regression is not possible, we have already provided an ablation study in Table 4, so please refer to this.
>
> There can be pros and cons to the approaches in SC-GS and SplineGS, but one sure thing is that our approach is computationally more efficient because it does not require any $K$-nearest neighbor search ( $O(n \log n)$ ). Performance-wise, the proposed approach is also quite effective as demonstrated in the experiments.
>
> ### Q4. Providing visualization videos.
> Here, we provide a video [(anonymous link)](https://anonymous.4open.science/r/SplineGS-0AF1/SplineGS_video.mp4) comparing the qualitative results of SplineGS and two of the existing methods, which can also be found in the new supplementary material. The video contains the visualizations for the D-NeRF and Neu3D datasets, which demonstrates that SplineGS can provide high-quality results with natural, realistic trajectories based on NURBS. Notably, SplineGS produces smooth trajectories even when increasing the rendering frame rate.
>
> ### Q5. Quantitative comparison on the HyperNeRF dataset.
> Here, we provide the results on the HyperNeRF dataset as requested (which has also been added to Table 9 in the appendix). SplineGS achieves comparable performance to the other methods. A critical problem with the HyperNeRF dataset is that the provided camera pose information is not accurate, which is why many recent works omit this dataset. An additional camera pose optimization is required to handle this problem, which is another set of problems left as future work.
>
> | Method   | 3D Printer | Chicken | Broom |
> |----------|------------|---------|-------|
> | D-3DGS   | 20.28      | 22.76   | 20.47 |
> | 4D-GS    | 22.10      | 28.70   | 22.00 |
> | SplineGS | 21.97      | 29.13   | 20.56 |
>
> ### Q6. Lack of some 4DGS metrics.
> These metrics were not reported in the original 4DGS paper; however, as shown in Table 1, the method takes 7.5 hours on average on the D-NeRF dataset. Thus, training on Neu3D is also expected to require a significant amount of time.

---

> ### Comment · Reviewer_x185 · 2024-11-24
>
> Thank you for your detailed rebuttal. Most of our concerns have been addressed.

---

> > ### Author Response · Authors · 2024-11-24
> >
> > We are pleased to hear that most of your concerns have been addressed. Your insightful questions have enabled us to clarify our contributions more clearly. If you find our revisions satisfactory, we kindly ask you to consider adjusting your rating accordingly.

---

### Official Review · Reviewer_AxDz · 2024-11-01

**Soundness:** 3
**Presentation:** 3
**Contribution:** 3
**Rating:** 6
**Confidence:** 4

**Summary:**

This paper proposed a new gaussian splatting method to reconstruct the dynamic scene. This paper investigate the usage of non-uniform rational B-splines (NURBS) in temporally smooth deformation representation. Based on the NURBS, they learned a set of trajectories and than linearly combine these trajectories to get the individual trajectories. They can achieve the competitive performance over the existing methods with much shorter training time.

**Strengths:**

1. This paper combined a new representation, called non-uniform rational B-splines (NURBS), with gaussian splatting to reconstruct the dynamic scene, which is declared to be effective in representing smooth deformation.
2. The results show that this method can achieve higher or competitive performance with existing SOTA methods in both the monocular dynamic and multi-view dynamic benchmarks.

**Weaknesses:**

1. To some content, this method still relies on the canonical space (the initial gaussians), but using different strategy to model the deformation. So how about the performance in the non-rigid scenes (scenes like HyperNeRF-data[1])?
2. Actually, there are some similar strategies or representations can achieve the same effect, like Fourier (Gaussion-flow [2] adopted this presentation). The crucial point is finding a more robust representation to model the deformation or trajectory. So what's the difference or advantage of the used NURBS over other representations like Fourier?

[1] HyperNeRF: A Higher-Dimensional Representation for Topologically Varying Neural Radiance Fields.
[2] Gaussian-Flow: 4D Reconstruction with Dynamic 3D Gaussian Particle.

**Questions:**

Compared with 4DGS, why this method has a significant advantage on D-NeRF dataset but has a relative small advantage on Neu3D dataset?

---

> ### Author Response · Authors · 2024-11-20
>
> ### W1. Quantitative comparison on the HyperNeRF dataset.
> Here, we provide the results on the HyperNeRF dataset as requested (which has also been added to Table 9 in the appendix). SplineGS achieves comparable performance to the other methods. A critical problem with the HyperNeRF dataset is that the provided camera pose information is not accurate, which is why many recent works omit this dataset. An additional camera pose optimization is required to handle this problem, which is another set of problems left as future work.
>
> | Method   | 3D Printer | Chicken | Broom |
> |----------|------------|---------|-------|
> | D-3DGS   | 20.28      | 22.76   | 20.47 |
> | 4D-GS    | 22.10      | 28.70   | 22.00 |
> | SplineGS | 21.97      | 29.13   | 20.56 |
>
> ### W2. Advantages of NURBS compared to Fourier representation.
> Fourier basis can also handle smooth changes, however, it is a global representation, i.e., changing its coefficients affects the entire trajectory. On the other hand, SplineGS can concisely represent temporally local changes. SplineGS requires only $p+1$ nearby control points to represent the deformation at a specific time. This is clear from Equation (5), where all but $p+1$ basis functions reduce to zero. This is also beneficial in rendering, i.e., saving computations for unnecessary control points. Real-world deformations are often composed of many temporally local deformations, so the proposed approach can be advantageous in this regard, especially in handling long video sequences. This discussion has been added in Section 2.2.
>
> ### Q1. Relatively smaller performance gap on Neu3D.
> The Neu3D dataset, consisting of real data, involves changes in color and opacity. Some models, such as 4DGS and STG, incorporate modeling for these. On the other hand, SplineGS concentrates only on the changes in blob positions and rotations, showing a relatively small performance gap for this dataset compared to that in the D-NeRF dataset. Nevertheless, SplineGS achieves higher performance despite having fewer deformation components, suggesting that the proposed representation is quite effective for trajectory modeling. This discussion was already present in Section 4. Considering color and density changes is a separate research direction that requires careful consideration, which is not the main focus of this paper.

---

> ### Comment · Reviewer_AxDz · 2024-11-24
>
> Thanks for authors' response. It seems that this method has the same problem on non-rigid scenes and is hard to solve the new-coming objects. If the explaination of 'Q1' is true that the 4DGS method additionally model the color and opacity, this method will make a great improvement once combined with these strategies. And maybe it's better to have a try to prove this and improve the model.

---

> > ### Author Response · Authors · 2024-11-24
> >
> > Thank you for your insightful comments. We agree that incorporating opacity and color models will improve the proposed method. We believe that we need to come up with new opacity and color models for SplineGS, considering that those of existing models are tailored to their own frameworks (e.g., using a global temporal basis for color in 4DGS, color handling in 2D space in STG, etc.). Following your suggestion, and as mentioned in Conclusion, we plan to address this as part of our future work.

---

### Official Review · Reviewer_cEb3 · 2024-11-04

**Soundness:** 3
**Presentation:** 3
**Contribution:** 3
**Rating:** 6
**Confidence:** 4

**Summary:**

This paper proposes SplineGS, a novel method for reconstructing dynamic scenes using 3D Gaussian splatting. The authors propose to utilize Non-Uniform Rational B-Splines (NURBS) to represent the trajectories of Gaussian blobs, aiming to achieve temporally smooth deformations. This approach is a departure from existing methods that rely on neural representations, which often require explicit regularization or specialized training mechanisms to ensure smooth trajectories. The paper demonstrates the effectiveness of SplineGS on two datasets, D-NeRF (synthetic) and Neu3D (real-world), showing competitive or superior performance compared to some existing methods.

**Strengths:**

The paper presents a novel approach by employing NURBS to represent Gaussian blob trajectories in dynamic scene reconstruction, a significant contribution to the field of 3D Gaussian splatting.

By utilizing NURBS, SplineGS provides smooth deformations without the need for explicit regularizers or complex training mechanisms, leading to faster and more efficient training. This efficiency translates to competitive or state-of-the-art performance on both synthetic (D-NeRF) and real-world (Neu3D) datasets, demonstrating its effectiveness in reconstructing dynamic scenes.

**Weaknesses:**

1. While the authors claim that NURBS inherently provide temporal smoothness, the paper lacks direct visual evidence to support this claim. The authors should include visualizations of the Gaussian trajectories generated by SplineGS, compared with trajectories from other methods. For instance, plotting the trajectories over time for a specific Gaussian blob in SplineGS and other methods like 4D-GS or D-3DGS would visually illustrate the differences in smoothness.

2. The paper focuses on modeling the deformation of Gaussian centers and rotations but omits incorporating scaling into its deformation model. This omission could limit the expressiveness and applicability of SplineGS, especially when dealing with objects that undergo significant changes in size or shape during motion. The authors should address this limitation and discuss its potential impact.

3. While the paper claims novelty in its use of NURBS for representing Gaussian blob trajectories, this claim requires further examination. The Spline-NeRF paper (Knodt, 2022) explores the use of Bezier splines for trajectory representation in dynamic scene reconstruction, a conceptually similar approach albeit within a NeRF framework. The authors should cite and acknowledge this related work and elaborate on the specific novelties of their NURBS-based method within the context of 3D Gaussian splatting.

4. The evaluation primarily focuses on datasets with limited camera motion. To comprehensively assess SplineGS's capabilities, the evaluation should be broadened to include datasets with more pronounced camera motion, such as HyperNeRF dataset. This would provide valuable insights into SplineGS's performance under challenging conditions and enable a more direct and thorough comparison with baseline methods like 4D-GS and Deformable 3D Gaussians, which are evaluated on HyperNeRF.

**Questions:**

See the Weaknesses section.

---

> ### Author Response · Authors · 2024-11-20
>
> ### W1. Visual comparison with existing methods.
> Here, we provide a video [(anonymous link)](https://anonymous.4open.science/r/SplineGS-0AF1/SplineGS_video.mp4) comparing the qualitative results of SplineGS and two of the existing methods, which can also be found in the new supplementary material. The video contains the renderings of the standup, jumping jacks, and mutant scenes from the D-NeRF datasets. Here, one can also find a comparison between the trajectories of SplineGS and D-3DGS. The video demonstrates the inherent smoothness of the NURBS trajectories in SplineGS. Especially, SplineGS produces smooth trajectories even when increasing the rendering frame rate. The video also presents rendering results and trajectory visualizations for the Neu3D dataset.
>
> ### W2. Omission of scale variation.
> It is true that the omission of scale variation prohibits some types of objects, e.g., jelly- or smoke-like objects. In reality, however, parts of an object exhibit minimal changes in scaling for most types of objects. It has also been reported in the literature that most non-rigid deformations are likely rigid transforms at infinitesimal levels [1]. This perspective is also adequate for Gaussian splatting, considering that the blobs represent some small parts forming the objects in the scene.
> Therefore, we concentrated on modeling the deformations of Gaussian centers and rotations. In reality, modeling scale variations tend to induce overfitting in our experience due to the additional degrees of freedom. Accordingly, we believe that scaling variations should be addressed with careful consideration as a separate research focus. We have added this discussion to Appendix A.
>
> [1] Lee et al., Consensus of non-rigid reconstructions, CVPR 2016.
>
> ### W3. The Spline-NeRF paper (Knodt, 2022).
> We thank the reviewer for this suggestion. We have added Spline-NeRF to the related work section and discussed the differences from our work. In fact, Spline-NeRF and SplineGS are structurally very different. Spline-NeRF takes all sampled points along a ray as inputs to an MLP, which then outputs control points for B&eacute;zier curves. In other words, the MLP's output defines the entire curve in Spline-NeRF. In contrast, our approach treats the control points for NURBS as learnable parameters and an MLP outputs the weights for representative trajectories, modeling spatial smoothness.
>
> Moreover, the B&eacute;zier curve used in Spline-NeRF is a global representation, i.e., changing one of the control points affects the entire curve. This has some downsides in modeling real-world deformations, where many temporally local deformations are likely combined. On the other hand, NURBS used in SplineGS requires only $p+1$ nearby control points to calculate the deformation at time $t$, allowing it to adapt flexibly to long video sequences.
>
> ### W4. Quantitative comparison on the HyperNeRF dataset.
> Here, we provide the results on the HyperNeRF dataset as requested (which has also been added to Table 9 in the appendix). SplineGS achieves comparable performance to the other methods. A critical problem with the HyperNeRF dataset is that the provided camera pose information is not accurate, which is why many recent works omit this dataset. An additional camera pose optimization is required to handle this problem, which is another set of problems left as future work.
>
> | Method   | 3D Printer | Chicken | Broom |
> |----------|------------|---------|-------|
> | D-3DGS   | 20.28      | 22.76   | 20.47 |
> | 4D-GS    | 22.10      | 28.70   | 22.00 |
> | SplineGS | 21.97      | 29.13   | 20.56 |

---

> ### Author Response · Authors · 2024-11-24
>
> Reviewer cEb3, we are still waiting for your response. We have addressed all your suggestions and questions in the rebuttal. We hope that our response clears all your concerns, possibly increasing the review score.

---

> > ### Comment · Reviewer_cEb3 · 2024-11-26
> >
> > Thank the authors' rebuttal. The authors have invested considerable effort in addressing the concerns raised, demonstrating a commitment to improving their work. They have provided visual evidence to support their claims about smoothness, justified the decision to exclude scale deformation, incorporated a discussion on Spline-NeRF, and included evaluations on the HyperNeRF dataset. Additionally, they have clarified their position on the omission of scale deformation, emphasizing that NURBS, 4DGS, and D-3DGS do not explicitly model scale variations. This clarification highlights that the focus of these methods is primarily on positional and rotational deformations, which effectively capture the dynamics of most non-rigid objects. However, certain responses, particularly regarding the limitations of excluding scale deformation and the challenges posed by the HyperNeRF dataset, could benefit from more detailed and convincing explanations. Further elaboration in these areas would strengthen the paper.
> >
> > While the authors have clarified that 4DGS and D-3DGS do not explicitly model scale variations, it is important to acknowledge that these methods may implicitly capture some degree of scaling through their respective deformation models. Therefore, it would be beneficial to provide a more in-depth analysis of the potential limitations of omitting scale deformation and discuss potential avenues for incorporating it in future work. This would address the reviewer's concerns and provide a more comprehensive understanding of the method's capabilities and potential areas for improvement.
> >
> > Considering that most of my concerns have been partially addressed, I would upgrade my rating to 6.

---

> > > ### Author Response · Authors · 2024-11-28
> > >
> > > We sincerely appreciate your thoughtful review and the recognition of our efforts to address your concerns. There seems to be a slight confusion regarding our response, though; we didn't say that 4DGS and D-3DGS do not have scale variations, and they do include these as suspected by the reviewer. What we explained was that scale variations, however, tend to overfit in our empirical experience. We agree with the reviewer's point that modeling scaling deformation could potentially improve performance if it is carefully handled with suitable physical models. We will pursue this goal in our future work.

---

### Author Response · Authors · 2024-11-20
**Common response to all reviewers**

We would like to express our sincere gratitude for the insightful comments provided by the reviewers. We have revised the paper considering the comments, mostly clarifying some possible misconceptions/misleading phrases and providing more results. Here are some general responses to frequently asked questions, and we will also provide more detailed answers to the individual reviewers.

[supplementary material link](https://anonymous.4open.science/r/SplineGS-0AF1/)

### ● A new visualization video.
We provide a video [(anonymous link)](https://anonymous.4open.science/r/SplineGS-0AF1/SplineGS_video.mp4) comparing the qualitative results of SplineGS and two of the existing methods, which can also be found in the new supplementary material. The video contains the visualizations for the D-NeRF and Neu3D datasets, which demonstrates that SplineGS can provide high-quality results with natural, realistic trajectories based on NURBS. Notably, SplineGS produces smooth trajectories even when increasing the rendering frame rate.


### ● Quantitative comparison on the HyperNeRF dataset.
We provide the results on the HyperNeRF dataset as requested by many reviewers (which has also been added to Table 9 in the appendix). Here, SplineGS achieves comparable performance to the other methods. A critical problem with the HyperNeRF dataset is that the provided camera pose information is not accurate, which is why many recent works omit this dataset. An additional camera pose optimization is required to handle this problem, which is another set of problems left as future work.

| Method   | 3D Printer | Chicken | Broom |
|----------|------------|---------|-------|
| D-3DGS   | 20.28      | 22.76   | 20.47 |
| 4D-GS    | 22.10      | 28.70   | 22.00 |
| SplineGS | 21.97      | 29.13   | 20.56 |

### ● Relatively smaller performance gap on Neu3D.
The Neu3D dataset, consisting of real data, involves changes in color and opacity. Some models, such as 4DGS and STG, incorporate modeling for these. On the other hand, SplineGS concentrates only on the changes in blob positions and rotations, showing a relatively small performance gap for this dataset compared to that in the D-NeRF dataset. Nevertheless, SplineGS achieves higher performance despite having fewer deformation components, suggesting that the proposed representation is quite effective for trajectory modeling. This discussion was already present in Section 4. Considering color and density changes is a separate research direction that requires careful consideration, which is not the main focus of this paper.

### ● LPIPS results.
We provide the LPIPS results as requested by many reviewers (which have also been added to Table 7 in the appendix). SplineGS achieves the best performance in most cases.

| Method   | Trex   | Jumping Jacks | Hell Warrior | Stand up | Bouncing Balls | Mutant | Hook   | Lego   | Mean   |
|----------|--------|---------------|--------------|----------|----------------|--------|--------|--------|--------|
| 4D-GS   | 0.0131 | 0.0128        | 0.0369       | 0.0074   | 0.0155         | 0.0167 | 0.0272 | 0.0382 | 0.0210 |
| D-3DGS  | 0.0098 | 0.0126        | 0.0234       | 0.0063   | 0.0093         | 0.0052 | 0.0144 | 0.0183 | 0.0124 |
| SC-GS    | 0.0119 | 0.0115        | 0.0280       | 0.0072   | 0.0216         | 0.0069 | 0.0139 | 0.0499 | 0.0119 |
| SplineGS | 0.0047 | 0.0065        | 0.0091       | 0.0027   | 0.0025         | 0.0019 | 0.0065 | 0.0298 | 0.0080 |

### ● A new scatter plot comparing efficiency and performance.
We provide a scatter plot [(anonymous link)](https://anonymous.4open.science/r/SplineGS-0AF1/Figure5.png) illustrating the processing time and the number of blobs for various methods, which has also been added to Figure 5 in the appendix. In the plot, we can confirm that SplineGS has a high average FPS even though it uses large numbers of Gaussian blobs compared to the other methods. This demonstrates the inherent efficiency of the proposed representation. In the plot, we can also confirm that SplineGS provides high performance with moderate training time compared to the existing methods.

---

### Meta-Review · Area_Chair_FAFL · 2024-12-21

**Metareview:**

This paper proposes a method for reconstructing dynamic scenes using 3D Gaussian splatting where Non-Uniform Rational B-Splines (NURBS) are utilized to represent the trajectories of Gaussian blobs, aiming to achieve temporally smooth deformations. Based on the NURBS, the proposed method learns a set of trajectories and then linearly combines these trajectories to get the individual trajectories. Employing NURBS to represent Gaussian blob trajectories in dynamic scene reconstruction should be highly appreciated. By utilizing NURBS, the method provides smooth deformations without the need for explicit regularizers or complex training mechanisms, leading to faster and more efficient training.  On the other hand, the reviewers raised concerns regarding unclear novelty against prior works, insufficient analysis of the experimental results, limitation of the proposed.  Advantages against 4D-GS, D-3GS, Fourier representation should be clearly argued with evidence. The performances are not consistent over evaluated datasets: D-NeRF and Neu3D, meaning significant improvement on D-NeRF while small one on Neu3D compared with 4DGS.  Evaluation on HyperNeRF which is with more pronounced camera motion is needed.  Not handling scale deformation need in-depth discussion.  The authors provided additional experiments and in-depth analysis on the results, and argued advantages/contributions of the paper compared some similar works. The authors’ rebuttal and following discussion between the authors and the reviewers have resolved most of the raised concerns, leading to positive consensus for the paper among the reviewers.  This paper should be accepted, accordingly.  AC suggests the authors to deeply argue the potential extension of the proposed method.  Since the deformation considered in the paper is limited to position and rotation changes, in-depth discussion to handle scale, color, and/or density changes will be beneficial to the community.

**Additional Comments On Reviewer Discussion:**

See above.

---

### Decision · Program_Chairs · 2025-01-22

Accept (Poster)